# CaDA: Cross-Problem Routing Solver with Constraint-Aware Dual-Attention

**Han Li** [*1]  **Fei Liu** [*2]  **Zhi Zheng** [1]  **Yu Zhang** [1]  **Zhenkun Wang** [1]

## Abstract

Vehicle routing problems (VRPs) are significant combinatorial optimization problems (COPs) holding substantial practical importance. Recently, neural combinatorial optimization (NCO), which involves training deep learning models on extensive data to learn vehicle routing heuristics, has emerged as a promising approach due to its efficiency and the reduced need for manual algorithm design. However, applying NCO across diverse real-world scenarios with various constraints necessitates cross-problem capabilities. Current cross-problem NCO methods for VRPs typically employ a constraint-unaware model, limiting their cross-problem performance. Furthermore, they rely solely on global connectivity, which fails to focus on key nodes and leads to inefficient representation learning. This paper introduces a Constraint-Aware Dual-Attention Model (CaDA), designed to address these limitations. CaDA incorporates a constraint prompt that efficiently represents different problem variants. Additionally, it features a dual-attention mechanism with a global branch for capturing broader graph-wide information and a sparse branch that selectively focuses on the key node connections. We comprehensively evaluate our model on 16 different VRPs and compare its performance against existing cross-problem VRP solvers. CaDA achieves state-of-the-art results across all tested VRPs. Our ablation study confirms that each component contributes to its cross-problem learning performance. The source code for CaDA is publicly available at https://github.com/CIAM-Group/CaDA.

*Equal contribution [1]Guangdong Provincial Key Laboratory of Fully Actuated System Control Theory and Technology, School of Automation and Intelligent Manufacturing, Southern University of Science and Technology, Shenzhen, China [2]Department of Computer Science, City University of Hong Kong, Hong Kong, China. Correspondence to: Zhenkun Wang <wangzhenkun90@gmail.com>.

*Proceedings of the 42$^{st}$ International Conference on Machine Learning*, Vancouver, Canada. PMLR 267, 2025. Copyright 2025 by the author(s).

## 1. Introduction

Vehicle routing problems (VRPs) involve optimizing transportation costs for a fleet of vehicles to meet all customers' demands while adhering to various constraints. Numerous studies have focused on VRPs due to their extensive real-world applications in transportation, logistics, and manufacturing (Cattaruzza et al., 2017; Rodrigue, 2020). Traditional methods for solving VRPs include exact solvers and heuristic methods. Exact solvers, however, struggle with the NP-hard nature of the problem, making them prohibitively expensive to implement. On the other hand, heuristic methods are more cost-effective and provide near-optimal solutions but require significant expert input in their design. Recently, learning-based neural solvers have gained considerable attention and have been successfully applied to VRPs (Bengio et al., 2021; Kool et al., 2019; Bogyrbayeva et al., 2024; Berto et al., 2025). These solvers train networks to learn a heuristic, reducing the need for extensive manual algorithm design and minimizing computational overhead.

Despite the promising performance of neural solvers on VRPs, the majority of existing works require training a model for each type of routing problem (Kwon et al., 2020; Drori et al., 2020; Duan et al., 2020; Gao et al., 2020; Cappart et al., 2021; Zhao et al., 2021; Kool et al., 2022; Tyasnurita et al., 2017; Berto et al., 2024a). Given the diversity of real-world vehicle routing problems with varying constraints (Tan & Yeh, 2021), developing a distinct model for each routing problem is costly and hinders practical application.

To tackle this challenge, recent efforts have been made to develop cross-problem learning methods that can solve multiple VRPs with a single model (Liu et al., 2024; Zhou et al., 2024a; Berto et al., 2024b; Lin et al., 2024). These cross-problem methods typically employ an encoder-decoder framework and are trained using reinforcement learning. For example, MTPOMO (Liu et al., 2024) jointly trains a unified model across five VRPs, each with one or two constraints, enabling zero-shot generalization to problems that feature combinations of these constraints. MVMoE (Zhou et al., 2024a) employs a mixture-of-experts (MoE) (Shazeer et al., 2017) structure in the feed-forward layer of a transformer-based model to enhance its cross-problem learning capacity. Furthermore, RouteFinder (Berto et al., 2024b) directly

trains and tests sixteen VRPs using a proposed unified re-inforcement learning (RL) environment, which enables the simultaneous handling of different VRPs in the same training batch. Additionally, RouteFinder leverages a modern transformer-based model structure (Dubey et al., 2024), along with global embeddings, to enhance performance.

Despite these advancements, existing cross-problem models remain unaware of constraints (Liu et al., 2024; Zhou et al., 2024a; Berto et al., 2024b). As different constraints significantly alter the feasible solution space, this oversight notably limits the models' capabilities in cross-problem applications. Furthermore, existing methods employ a transformer encoder which maintains global connectivity throughout the node encoding process, leading to the inclusion of irrelevant nodes and adversely affecting node representation.

This study proposes a novel Constraint-Aware Dual-Attention Model (CaDA) to mitigate these challenges. Firstly, we introduce a constraint prompt to enhance the model's awareness of the activated constraints. Furthermore, we propose a dual-attention mechanism consisting of a global branch and a sparse branch. Since, in the encoder-decoder framework, node pairs with higher attention scores are more likely to be adjacent in the solution, the sparse branch with Top-$k$ sparse attention focuses on the more promising connections between these key node pairs. Meanwhile, the global branch enhances the model's capacity by capturing information from the entire graph, ensuring that the solution is informed globally. The effectiveness and superiority of CaDA have been comprehensively demonstrated across 16 VRPs and real-world benchmarks.

The contributions of this paper are as follows:

- We introduce CaDA, an efficient cross-problem learning method for VRPs that enhances model awareness of constraints and representation learning.

- We propose a constraint prompt, which facilitates high-quality constraint-aware learning, and a dual-attention mechanism, which ensures the encoding process is both selectively focused and globally informed.

- We conduct a comprehensive evaluation of CaDA across 16 VRP variants. CaDA achieves state-of-the-art (SOTA) performance, surpassing existing cross-problem learning methods. Additionally, our ablation study validates the effectiveness of both the constraint prompt and the dual-attention mechanism.

## 2. Preliminaries

### 2.1. Problem Definition

In this study, we focus on 16 VRP variants that encompass five different constraints, including capacity (C), open route

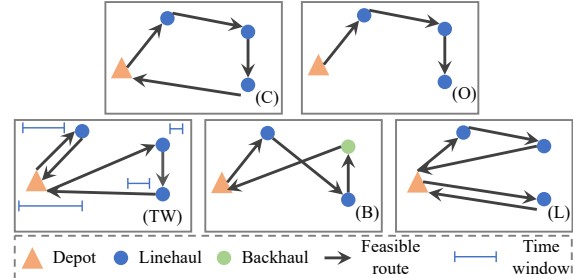

*Figure 1.* The feasible solutions for different VRP variants.

(O), backhaul (B), duration limit (L), and time window (TW). The illustrations of the constraints are presented in Figure 1. In this section, we begin by outlining a general definition of the VRP instance, then introduce the basic CVRP, and proceed to describe four additional constraints.

A VRP instance $\mathcal{G}$ is a fully-connected graph defined by a set of nodes $\mathcal{V} = \{v_0, v_1, \ldots, v_N\}$ with the total number of nodes given by $|\mathcal{V}| = N + 1$, and edges $\mathcal{E} = \mathcal{V} \times \mathcal{V}$. Furthermore, $v_0$ represents the depot, while $\{v_1, \ldots, v_N\}$ represent the $N$ customer nodes. Each node $v_i \in \mathcal{V}$ consists of the pair $\{\vec{X}_i, A_i\}$, where $\vec{X}_i \sim U(0, 1)^2$ represents the node coordinates, and $A_i$ denotes other attributes of the nodes. Additionally, the travel cost between different nodes is defined by their Euclidean distance, which is denoted by the cost matrix $\mathbf{D} = \{d_{i,j}, i = 0, \ldots, N, j = 0 \ldots, N\}$.

In CVRP, the depot node $v_0$ has $A_0 = \emptyset$, and each customer node $v_i$ is associated with $A_i = \{\delta_i\}$, where $\delta_i$ is the customer's demand at $v_i$ that the fleet of vehicles must service. This fleet comprises homogeneous vehicles, each with a specific capacity $C$. Each vehicle leaves the depot $v_0$, visits a subset of customers, and returns to the depot upon completion of deliveries. The solution to CVRP consists of the routes taken by all vehicles, i.e., $\{\boldsymbol{\sigma}^1, \boldsymbol{\sigma}^2, \ldots, \boldsymbol{\sigma}^K\}$, where $K$ is the total number of sub-routes. Each sub-route $\boldsymbol{\sigma}^k = (\sigma_1^k, \sigma_2^k, \ldots, \sigma_{n_k}^k)$, $k \in \{1, 2, \ldots, K\}$, where $\sigma_i^k$ is the index of the visited node at step $i$, and $\sigma_1^k = \sigma_{n_k}^k = 0$. $n_k = |\boldsymbol{\sigma}^k|$ represents the number of nodes in it, and $\sum_{k=1}^K n_k = T$ is the total number of visit steps.

The basic CVRP could be easily extended to various VRPs by adding additional constraints. This study explores four additional constraints as discussed in recent studies (Liu et al., 2024; Zhou et al., 2024a; Berto et al., 2024b).

**Open Route (O)** In the OVRP, vehicles do not return to the depot $v_0$ after completing their sub-route.

**Time Window (TW)** This constraint requires that each node must be visited within a specific time window, such that each node $A_i = \{\delta_i, e_i, l_i, s_i\}$, where $e_i$ is the earliest start time, $l_i$ is the latest permissible time, and $s_i$ represents the time taken to service this customer. The depot $v_0$ has

$s_0 = 0$, $e_0 = 0$, and $l_0 = \mathcal{T}$, indicating that each sub-tour must be completed within a time limit of $\mathcal{T}$. Time window constraints are stringent; if a vehicle arrives earlier than $e_i$, it must wait until the start of the window.

**Backhaul (B)** Customers with $\delta_i > 0$ are linehaul customers, requiring vehicles to load goods at the depot and deliver them to their locations, while those with $\delta_i < 0$ are backhaul customers, where vehicles collect $|\delta_i|$ goods from customers and return them to the depot. While all customers in the standard CVRP are linehaul, the VRP with backhauls (VRPB) includes types of customers, and linehaul tasks must precede backhaul tasks to avoid reloading.

**Duration Limit (L)** In this constraint, the depot $v_0$ has $A_0 = \{\rho\}$, where $\rho$ is the length limit that each sub-tour must adhere to.

### 2.2. Learning to Construct Solutions for VRPs

The process of constructing solutions autoregressively (i.e., decoding) can be modeled as a Markov decision process (MDP), and the policy can be trained using RL methods. As the model sequentially expands each sub-route, for simplicity, at any decoding step $t$, $\boldsymbol{\tau}_t$ represents the sequence of nodes visited up to that point:

$$\boldsymbol{\tau}_t = \bigcup_{k=1}^{K_t} (\tau_1^k, \tau_2^k, \ldots, \tau_{n_k}^k) = (\tau_1, \tau_2, \ldots, \tau_t), \quad (1)$$

where $\bigcup$ denotes the concatenation of sequences from different sub-routes, $K_t$ denotes the number of sub-tours up to the current step. The MDP for the decoding step at time $t$ is defined in the Appendix B.1.

Subsequently, policy $\pi_\theta$ can be optimized using RL methods to maximize the expected reward $J$. This study employs the REINFORCE algorithm (Williams, 1992) with a shared baseline proposed by Kwon et al. (2020), to update the policy. Specifically, for a VRP instance $\mathcal{V}$, $N$ trajectories are generated, starting with the first action $\{a_1^1, a_1^2, \ldots, a_1^N\}$, which is always 0. Each of the $N$ trajectories then assigns a unique one of the $N$ customer nodes as the second point, i.e., $\{a_2^1, a_2^2, \ldots, a_2^N\} = \{1, 2, \ldots, N\}$. The policy subsequently samples actions for each trajectory until all have derived feasible solutions $\{\boldsymbol{\tau}^1, \boldsymbol{\tau}^2, \ldots, \boldsymbol{\tau}^N\}$. Finally, the gradient of the policy is approximated by:

$$\nabla_\theta J(\theta \mid \mathcal{V}) \approx \frac{1}{N} \sum_{i=1}^{N} (r(\boldsymbol{\tau}^i) - b^i(\mathcal{V})) \nabla_\theta \log \pi_\theta(\boldsymbol{\tau}^i \mid \mathcal{V}),$$

$$b^i(\mathcal{V}) = \frac{1}{N} \sum_{j=1}^{N} r(\boldsymbol{\tau}^j) \quad \text{for all } i. \quad (2)$$

Where $b(\mathcal{V})$ is the shared baseline function used to stabilize learning, $r(\cdot)$ is the reward function and is defined as the negative solution length, as detailed in Appendix B.1.

For the structure of policy $\pi_\theta$, existing approaches primarily use transformer-based models.

### 2.3. Transformer Layer

The Transformer (Vaswani et al., 2017) comprises a multi-head attention layer (MHA) and a feed-forward layer (FFD). In some modern large language models (Chowdhery et al., 2023; Touvron et al., 2023; Naveed et al., 2023), the FFD is replaced by gated linear units (GLUs), with a detailed introduction to GLUs provided in the Appendix B.2.

**Attention Layer** The classical attention function is:

$$\text{Attention}(X, Y) = \mathbf{A}(YW_V),$$
$$\text{where } \mathbf{A} = \text{Softmax}\left(\frac{XW_Q(YW_K)^\top}{\sqrt{d_k}}\right), \quad (3)$$

where $X \in \mathbb{R}^{n \times d}$ and $Y \in R^{m \times d}$ represent the input embeddings. The parameters $W_Q, W_K \in \mathbb{R}^{d \times d_k}$, and $W_V \in \mathbb{R}^{d \times d_v}$ are trainable matrices for the query, key, and value projections, respectively. After calculating the attention matrix using the query and key matrices, the Softmax function is applied independently across each row to normalize the attention scores. These scores are then rescaled by $\sqrt{d_k}$, resulting in the scaled attention score matrix $\mathbf{A}$. The eventual output, denoted as $Z$, is a matrix in $\mathbb{R}^{n \times d_v}$.

Additionally, for efficiency, the MHA projects $X$ into $M_h$ separate sets of queries, keys, and values, upon which the attention function is applied:

$$\text{MHA}(X, Y) = \text{Concat}(Z_1, \ldots, Z_{M_h})W_p,$$
$$\text{where} \quad Z_i = \text{Attention}_i(X, Y), \forall i \in \{1, \ldots, M_h\}, \quad (4)$$

where $d_k = d_v = \frac{d}{M_h}$ in each $\text{Attention}_i$. $W_p \in \mathbb{R}^{d \times d}$ is a learnable parameter. For self-attention, we have $Y = X$.

## 3. Methodology

### 3.1. Overall Pipeline

As shown in Figure 2, CaDA follows the general cross-problem learning framework for VRPs which consists of two stages: encoding the instance $\mathcal{V}$ to node embeddings $H^{(L)}$, and decoding to construct solutions based on $H^{(L)}$ sequentially. CaDA employs a prompt to introduce constraint information during the encoding process, so that the encoder can identify whether certain constraints are valid. In addition, CaDA utilizes a dual-attention mechanism. The sparse branch uses Top-$k$ sparse attention to concentrate on the most promising adjacent node candidates. This enables

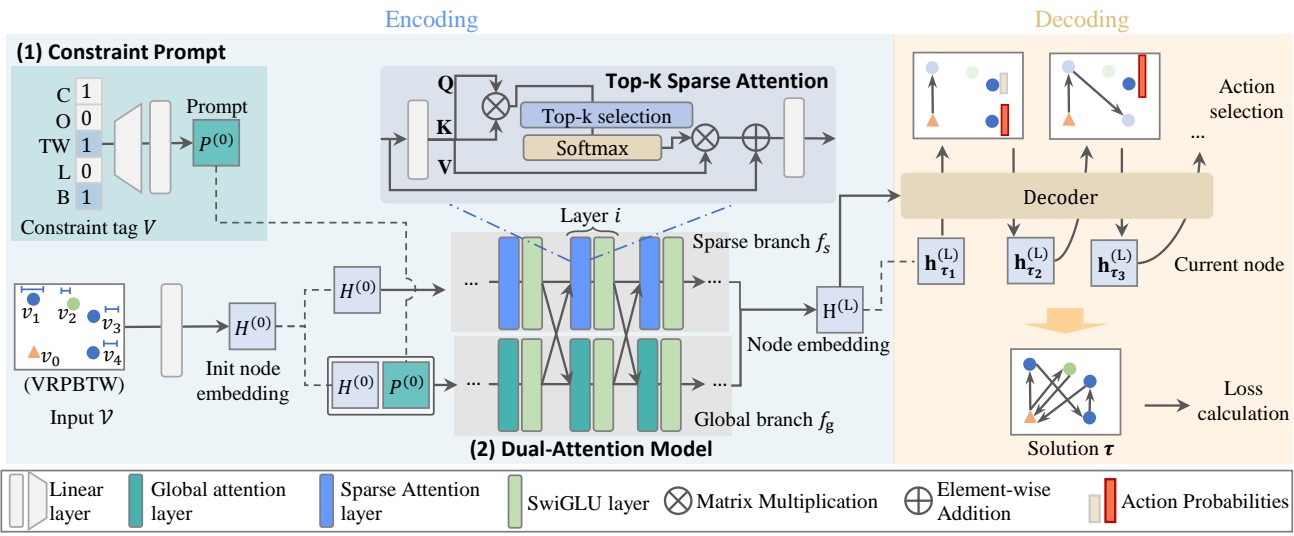

*Figure 2.* The pipeline of the proposed CaDA for VRPs. CaDA adopts the typical encoder-decoder framework and incorporates two new components in the encoder: a dual-attention mechanism and a constraint prompt. The dual-attention mechanism comprises a global branch with the standard Softmax function and a sparse branch with Top-$k$ selection operation.

the model to identify and focus on highly relevant node pairs based on learnable attention scores.

### 3.2. Constraint Prompt

To generate prompts that carry the problem's constraint information, we represent the problem as a multi-hot vector $V \in \mathbb{R}^5$, corresponding to five distinct constraints. This multi-hot vector is subsequently processed through a straightforward multi-layer perceptron (MLP) to generate the prompts:

$$P^{(0)} = \text{LayerNorm}(V W_a + b_a) W_b + b_b, \quad (5)$$

where $W_a \in \mathbb{R}^{5 \times d_h}$, $b_a \in \mathbb{R}^{d_h}$, $W_b \in \mathbb{R}^{d_h \times d_h}$, and $b_b \in \mathbb{R}^{d_h}$ are learnable parameters. $d_h$ is the node's embedding dimension. Then this prompt can be concatenated with the node embeddings.

### 3.3. Dual-Attention Mechanism

The input instance $\mathcal{V}$ with $|\mathcal{V}| = N + 1$, is first transformed into high-dimensional initial node embeddings by a linear projection. The initial node embedding is denoted as $H^{(0)} \in \mathbb{R}^{(N+1) \times d_h}$.

Subsequently, $H^{(0)}$ is concatenated with $P^{(0)}$ and processed through a global branch $f_g$, which consists of $L$ layers. Each consists of a standard MHA layer (Vaswani et al., 2017) and a SwiGLU (Shazeer, 2020). The standard attention function with Softmax never allocates exactly zero weight to any node, thereby allowing each node access to the entire graph. Concurrently, to capture information from closely related nodes, a sparse branch denoted as $f_s$ with Top-$k$ sparse attention layers is introduced. Both branches adaptively

fuse information at the end of each layer.

Finally, the output from the global branch, $H_g^{(L)}$, is used for autoregressive decoding, with the likelihood of node selection being primarily determined by the similarity of the nodes' embeddings.

**Global Layer** Each layer involves an MHA (Vaswani et al., 2017) and a SwiGLU (Shazeer, 2020), along with root mean square normalization (RMSNorm) (Zhang & Sennrich, 2019) and residual connections (He et al., 2016). Following Berto et al. (2024b), we employ SwiGLU and RMSNorm to improve convergence. The $i$-th layer is formulated as follows:

$$\hat{H}_g^{(i)} = \text{RMSNorm}^{(i)} \left( H_g^{(i-1)} + \text{MHA}^{(i)} \right.$$
$$\left. \left( H_g^{(i-1)}, \text{Concat}\left[ H_g^{(i-1)}, P^{(i-1)} \right] \right) \right), \quad (6)$$

$$\tilde{H}_g^{(i)} = \text{RMSNorm}^{(i)} \left( \hat{H}_g^{(i)} + \text{SwiGLU}^{(i)}(\hat{H}_g^{(i)}) \right), \quad (7)$$

$$\hat{P}^{(i)} = \text{RMSNorm}^{(i)} \left( P^{(i-1)} + \text{MHA}^{(i)} \right.$$
$$\left. \left( P^{(i-1)}, \text{Concat}\left[ H_g^{(i-1)}, P^{(i-1)} \right] \right) \right), \quad (8)$$

$$P^{(i)} = \text{RMSNorm}^{(i)} \left( \hat{P}^{(i)} + \text{SwiGLU}^{(i)}(\hat{P}^{(i)}) \right), \quad (9)$$

where $H_g^{(i-1)} \in \mathbb{R}^{(N+1) \times d_h}$ represents the node embeddings output from the $(i-1)$-th global layer.

**Sparse Layer** In the sparse branch $f_s$, each layer also consists of an attention layer and a SwiGLU activation function. However, to focus more precisely on related nodes, we replace the attention function $\text{Attention}(\cdot, \cdot)$ in $\text{MHA}(\cdot, \cdot)$

with SparseAtt$(\cdot, \cdot)$, which masks attention scores smaller than the Top-$k$ scores by setting them to zero. This can be formulated as follows:

$$\text{SparseAtt}(X, Y) = \text{Softmax}\left(M(\mathbf{A})\right) Y W_V, \quad (10)$$

where $\mathbf{A}$ is the attention score calculated as shown in Equation 3. $M(\cdot)$ is the Top-$k$ selection operation:

$$[M(\mathbf{A})]_{ij} = \begin{cases} \mathbf{A}_{ij} & \text{if } \mathbf{A}_{ij} \in \text{Top-}k(\mathbf{A}_{i*}), \\ 0 & \text{otherwise.} \end{cases} \quad (11)$$

where $\mathbf{A}_{i*}$ represents the attention scores of the $i$-th node with all other nodes, i.e., $\mathbf{A}_{i*} = \{\mathbf{A}_{ij} \mid j \in \{0, 1, \ldots, N\}\}$, and the Top-$k$ operation selects the top $k$ highest attention scores from this set.

**Fusion Layer**   In our model, a simple linear projection is applied at the end of each layer to transform embeddings between two branches. For the $i$-th layer, the outputs from the global and sparse branches are denoted as $\tilde{H}_g^{(i)}$ and $\tilde{H}_s^{(i)}$, respectively. The final outputs are given by:

$$H_g^{(i)} = \tilde{H}_g^{(i)} + (\tilde{H}_s^{(i)} W_s + b_s), \quad (12)$$

$$H_s^{(i)} = \tilde{H}_s^{(i)} + (\tilde{H}_g^{(i)} W_g + b_g), \quad (13)$$

where $W_s$, $b_s$, $W_g$, and $b_g$ are learnable parameters.

## 3.4. Decoder

After encoding, the output of the global branch, $H^{(L)} = [\boldsymbol{h}_0^{(L)}, \boldsymbol{h}_1^{(L)}, \ldots, \boldsymbol{h}_N^{(L)}]$, is utilized to construct the solution. During the autoregressive decoding process, at step $t$, the context embedding is defined as:

$$H_c = \text{Concat}\left[\boldsymbol{h}_{\tau_t}^{(L)}, c_t^{\text{l}}, c_t^{\text{b}}, z_t, l_t, o_t\right] W_t, \quad (14)$$

where $\tau_t$ is the partial solution already generated, and $\tau_t$ is the last node of the partial solution. The terms $c_t^{\text{l}}, c_t^{\text{b}}$ represent the remaining capacity of the vehicle for linehaul and backhaul customers, respectively. The terms $z_t$, $l_t$, and $o_t$ represent the current time, the remaining length of the current partial route (if the problem includes a length limitation), and the presence indicator of the open route, respectively. The matrix $W_c \in \mathbb{R}^{(d_h+5) \times d_h}$ is a learnable parameter.

Then the context embeddings are processed through an MHA to generate the final query:

$$q_c = \text{MHA}(H_c, \text{Concat}\left[\boldsymbol{h}_i^{(L)} : i \in I_t\right]), \quad (15)$$

where $I_t$ represents the set of feasible actions at the current step. The compatibility $u_i$ is computed as:

$$u_i = \begin{cases} \xi \cdot \tanh\left(\frac{q_c (\boldsymbol{h}_i^{(L)})^\top}{\sqrt{d_k}}\right) & \text{if } i \in I_t, \\ -\infty & \text{otherwise,} \end{cases} \quad (16)$$

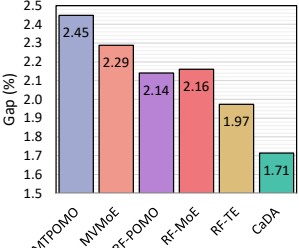 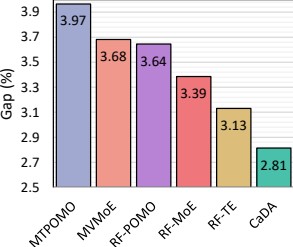

(a) Average gap of different neural solvers on VRPs with 50 nodes.   (b) Average gap of different neural solvers on VRPs with 100 nodes.

*Figure 3.* Comparison results of CaDA with SOTA cross-problem neural solvers, showing the average gap on 16 VRPs.

where $\xi$ is a predefined clipping hyperparameter. Finally, the action probabilities $\pi_\theta(\tau_t = i \mid \mathcal{V}, \boldsymbol{\tau}_{1:t-1})$ are obtained by applying the Softmax function to $\mathbf{u} = \{u_i\}_{i \in I}$. Additionally, the feasibility testing process for determining the set of feasible actions $I_t$ at the current step is detailed in Appendix B.3.

## 4. Experiments

To evaluate the effectiveness of the proposed CaDA for VRPs, we conduct experiments on 16 VRP variants with five constraints. Furthermore, we perform ablation studies to validate the efficiency of the proposed components.

## 4.1. Baselines

We utilize SOTA traditional and neural solvers as baselines. For the traditional solvers, we use PyVRP (Wouda et al., 2024), an extension of HGS-CVRP (Vidal, 2022), and Google's OR-Tools. Both baselines run on a single CPU core with time limits of 10s for VRP50 and 20s for VRP100. For the neural solvers, we compare our method against representative multi-task learning models: MT-POMO (Liu et al., 2024), MVMoE (Zhou et al., 2024a), and RouteFinder (Berto et al., 2024b), including RF-POMO, RF-MoE, and RF-TE. We utilize the open-source code published by RouteFinder (Berto et al., 2024b). For each method, we train two models from scratch on VRP50 and VRP100 using the same hyperparameters and problem settings as in Berto et al. (2024b). The detailed problem setup can be found in Appendix C.1. However, for RF-MoE and MVMoE, due to their higher memory demands, we utilize the pre-trained parameters provided by RouteFinder (Berto et al., 2024b) and test them under the same hardware settings as ours. All neural methods are trained using the same data budget. For CaDA, we train two models on VRP50 and VRP100, with the hyperparameters outlined in Appendix C.2. Mixed-batch training is employed to stabilize the training process (Berto et al., 2024b). Among the neural solvers, MVMoE has the largest model size (3.7 M), followed by CaDA (3.4 M),

*Table 1.* Performance on 1K test instances of 16 VRPs. The best learning-based results are highlighted with a gray background.

| Problem | Solver | n=50 Obj. | n=50 Gap | n=50 Time | n=100 Obj. | n=100 Gap | n=100 Time |
|---|---|---|---|---|---|---|---|
| CVRP | HGS-PyVRP | 10.372 | * | 10.4m | 15.628 | * | 20.8m |
| | OR-Tools | 10.572 | 1.907% | 10.4m | 16.280 | 4.178% | 20.8m |
| | MTPOMO | 10.520 | 1.423% | 2s | 15.941 | 2.030% | 8s |
| | MVMoE | 10.499 | 1.229% | 3s | 15.888 | 1.693% | 11s |
| | RF-POMO | 10.506 | 1.300% | 2s | 15.908 | 1.833% | 8s |
| | RF-MoE | 10.499 | 1.225% | 3s | 15.877 | 1.625% | 11s |
| | RF-TE | 10.502 | 1.257% | 2s | 15.860 | 1.524% | 8s |
| | CaDA | 10.494 | 1.182% | 2s | 15.870 | 1.578% | 8s |
| OVRP | HGS-PyVRP | 6.507 | * | 10.4m | 9.725 | * | 20.8m |
| | OR-Tools | 6.553 | 0.686% | 10.4m | 9.995 | 2.732% | 20.8m |
| | MTPOMO | 6.717 | 3.194% | 2s | 10.216 | 5.028% | 8s |
| | MVMoE | 6.705 | 3.003% | 3s | 10.177 | 4.617% | 11s |
| | RF-POMO | 6.699 | 2.926% | 2s | 10.190 | 4.761% | 8s |
| | RF-MoE | 6.697 | 2.880% | 3s | 10.139 | 4.234% | 11s |
| | RF-TE | 6.682 | 2.658% | 2s | 10.115 | 3.996% | 8s |
| | CaDA | 6.670 | 2.468% | 2s | 10.121 | 4.045% | 8s |
| VRPB | HGS-PyVRP | 9.687 | * | 10.4m | 14.377 | * | 20.8m |
| | OR-Tools | 9.802 | 1.159% | 10.4m | 14.933 | 3.853% | 20.8m |
| | MTPOMO | 10.036 | 3.596% | 2s | 15.102 | 5.052% | 8s |
| | MVMoE | 10.007 | 3.292% | 3s | 15.023 | 4.505% | 10s |
| | RF-POMO | 9.992 | 3.135% | 2s | 15.025 | 4.534% | 8s |
| | RF-MoE | 9.980 | 3.017% | 3s | 14.973 | 4.168% | 10s |
| | RF-TE | 9.979 | 3.000% | 2s | 14.935 | 3.906% | 8s |
| | CaDA | 9.960 | 2.800% | 2s | 14.960 | 4.038% | 8s |
| VRPBL | HGS-PyVRP | 10.186 | * | 10.4m | 14.779 | * | 20.8m |
| | OR-Tools | 10.331 | 1.390% | 10.4m | 15.426 | 4.338% | 20.8m |
| | MTPOMO | 10.679 | 4.760% | 2s | 15.718 | 6.294% | 8s |
| | MVMoE | 10.639 | 4.384% | 3s | 15.642 | 5.771% | 11s |
| | RF-POMO | 10.590 | 3.926% | 2s | 15.632 | 5.725% | 8s |
| | RF-MoE | 10.575 | 3.765% | 3s | 15.542 | 5.125% | 10s |
| | RF-TE | 10.569 | 3.713% | 2s | 15.523 | 5.008% | 8s |
| | CaDA | 10.543 | 3.461% | 2s | 15.525 | 5.001% | 8s |
| VRPBTW | HGS-PyVRP | 18.292 | * | 10.4m | 29.467 | * | 20.8m |
| | OR-Tools | 18.366 | 0.383% | 10.4m | 29.945 | 1.597% | 20.8m |
| | MTPOMO | 18.649 | 1.938% | 2s | 30.478 | 3.426% | 9s |
| | MVMoE | 18.632 | 1.841% | 3s | 30.437 | 3.284% | 12s |
| | RF-POMO | 18.603 | 1.684% | 2s | 30.384 | 3.102% | 9s |
| | RF-MoE | 18.616 | 1.757% | 3s | 30.340 | 2.951% | 12s |
| | RF-TE | 18.573 | 1.517% | 2s | 30.249 | 2.641% | 9s |
| | CaDA | 18.500 | 1.117% | 2s | 30.059 | 1.999% | 9s |
| OVRPB | HGS-PyVRP | 6.898 | * | 10.4m | 10.335 | * | 20.8m |
| | OR-Tools | 6.928 | 0.412% | 10.4m | 10.577 | 2.315% | 20.8m |
| | MTPOMO | 7.105 | 2.973% | 2s | 10.882 | 5.264% | 8s |
| | MVMoE | 7.089 | 2.744% | 3s | 10.841 | 4.869% | 11s |
| | RF-POMO | 7.085 | 2.686% | 2s | 10.839 | 4.857% | 8s |
| | RF-MoE | 7.081 | 2.617% | 3s | 10.806 | 4.528% | 11s |
| | RF-TE | 7.065 | 2.385% | 2s | 10.774 | 4.233% | 8s |
| | CaDA | 7.049 | 2.159% | 2s | 10.762 | 4.099% | 8s |
| OVRPBLTW | HGS-PyVRP | 11.668 | * | 10.4m | 19.156 | * | 20.8m |
| | OR-Tools | 11.681 | 0.106% | 10.4m | 19.305 | 0.767% | 20.8m |
| | MTPOMO | 11.823 | 1.315% | 3s | 19.658 | 2.602% | 9s |
| | MVMoE | 11.816 | 1.249% | 4s | 19.640 | 2.514% | 12s |
| | RF-POMO | 11.810 | 1.192% | 3s | 19.618 | 2.393% | 10s |
| | RF-MoE | 11.824 | 1.309% | 4s | 19.607 | 2.334% | 12s |
| | RF-TE | 11.789 | 1.017% | 2s | 19.554 | 2.061% | 9s |
| | CaDA | 11.760 | 0.771% | 2s | 19.435 | 1.439% | 9s |
| OVRPL | HGS-PyVRP | 6.507 | * | 10.4m | 9.724 | * | 20.8m |
| | OR-Tools | 6.552 | 0.668% | 10.4m | 10.001 | 2.791% | 20.8m |
| | MTPOMO | 6.720 | 3.248% | 2s | 10.224 | 5.112% | 8s |
| | MVMoE | 6.706 | 3.028% | 3s | 10.184 | 4.693% | 11s |
| | RF-POMO | 6.701 | 2.944% | 2s | 10.190 | 4.762% | 8s |
| | RF-MoE | 6.695 | 2.859% | 3s | 10.140 | 4.252% | 11s |
| | RF-TE | 6.683 | 2.680% | 2s | 10.121 | 4.054% | 8s |
| | CaDA | 6.671 | 2.475% | 2s | 10.122 | 4.052% | 8s |

| Problem | Solver | n=50 Obj. | n=50 Gap | n=50 Time | n=100 Obj. | n=100 Gap | n=100 Time |
|---|---|---|---|---|---|---|---|
| VRPTW | HGS-PyVRP | 16.031 | * | 10.4m | 25.423 | * | 20.8m |
| | OR-Tools | 16.089 | 0.347% | 10.4m | 25.814 | 1.506% | 20.8m |
| | MTPOMO | 16.419 | 2.423% | 2s | 26.433 | 3.962% | 9s |
| | MVMoE | 16.400 | 2.298% | 3s | 26.390 | 3.789% | 11s |
| | RF-POMO | 16.363 | 2.066% | 2s | 26.361 | 3.675% | 9s |
| | RF-MoE | 16.389 | 2.232% | 3s | 26.321 | 3.516% | 11s |
| | RF-TE | 16.341 | 1.933% | 2s | 26.228 | 3.154% | 8s |
| | CaDA | 16.278 | 1.536% | 2s | 26.070 | 2.530% | 8s |
| VRPL | HGS-PyVRP | 10.587 | * | 10.4m | 15.766 | * | 20.8m |
| | OR-Tools | 10.570 | 2.343% | 10.4m | 16.466 | 5.302% | 20.8m |
| | MTPOMO | 10.775 | 1.733% | 2s | 16.157 | 2.483% | 8s |
| | MVMoE | 10.753 | 1.525% | 3s | 16.099 | 2.113% | 11s |
| | RF-POMO | 10.748 | 1.498% | 2s | 16.117 | 2.241% | 8s |
| | RF-MoE | 10.737 | 1.390% | 3s | 16.070 | 1.937% | 11s |
| | RF-TE | 10.747 | 1.485% | 2s | 16.057 | 1.858% | 8s |
| | CaDA | 10.731 | 1.333% | 2s | 16.057 | 1.847% | 8s |
| OVRPTW | HGS-PyVRP | 10.510 | * | 10.4m | 16.926 | * | 20.8m |
| | OR-Tools | 10.519 | 0.078% | 10.4m | 17.027 | 0.583% | 20.8m |
| | MTPOMO | 10.676 | 1.558% | 2s | 17.442 | 3.022% | 9s |
| | MVMoE | 10.674 | 1.541% | 3s | 17.416 | 2.870% | 12s |
| | RF-POMO | 10.656 | 1.361% | 2s | 17.405 | 2.809% | 9s |
| | RF-MoE | 10.674 | 1.540% | 3s | 17.388 | 2.704% | 12s |
| | RF-TE | 10.645 | 1.264% | 2s | 17.328 | 2.352% | 9s |
| | CaDA | 10.613 | 0.957% | 2s | 17.226 | 1.751% | 9s |
| VRPBLTW | HGS-PyVRP | 18.361 | * | 10.4m | 29.026 | * | 20.8m |
| | OR-Tools | 18.422 | 0.332% | 10.4m | 29.830 | 2.770% | 20.8m |
| | MTPOMO | 19.001 | 2.199% | 3s | 30.948 | 3.794% | 9s |
| | MVMoE | 18.983 | 2.097% | 3s | 30.892 | 3.609% | 12s |
| | RF-POMO | 18.938 | 1.863% | 2s | 30.847 | 3.452% | 9s |
| | RF-MoE | 18.957 | 1.960% | 3s | 30.809 | 3.325% | 12s |
| | RF-TE | 18.910 | 1.713% | 2s | 30.705 | 2.978% | 9s |
| | CaDA | 18.848 | 1.376% | 2s | 30.520 | 2.359% | 9s |
| VRPLTW | HGS-PyVRP | 16.356 | * | 10.4m | 25.757 | * | 20.8m |
| | OR-Tools | 16.441 | 0.499% | 10.4m | 26.259 | 1.899% | 20.8m |
| | MTPOMO | 16.832 | 2.877% | 2s | 26.913 | 4.455% | 9s |
| | MVMoE | 16.817 | 2.783% | 3s | 26.866 | 4.272% | 12s |
| | RF-POMO | 16.756 | 2.419% | 2s | 26.818 | 4.084% | 9s |
| | RF-MoE | 16.777 | 2.548% | 3s | 26.773 | 3.910% | 12s |
| | RF-TE | 16.728 | 2.248% | 2s | 26.706 | 3.645% | 9s |
| | CaDA | 16.669 | 1.879% | 2s | 26.540 | 2.995% | 9s |
| OVRPBL | HGS-PyVRP | 6.899 | * | 10.4m | 10.335 | * | 20.8m |
| | OR-Tools | 6.927 | 0.386% | 10.4m | 10.582 | 2.363% | 20.8m |
| | MTPOMO | 7.112 | 3.053% | 2s | 10.888 | 5.318% | 8s |
| | MVMoE | 7.094 | 2.799% | 3s | 10.847 | 4.929% | 11s |
| | RF-POMO | 7.088 | 2.703% | 2s | 10.842 | 4.883% | 8s |
| | RF-MoE | 7.082 | 2.630% | 3s | 10.807 | 4.537% | 11s |
| | RF-TE | 7.068 | 2.417% | 2s | 10.778 | 4.266% | 8s |
| | CaDA | 7.051 | 2.166% | 2s | 10.762 | 4.102% | 8s |
| OVRPBTW | HGS-PyVRP | 11.669 | * | 10.4m | 19.156 | * | 20.8m |
| | OR-Tools | 11.682 | 0.109% | 10.4m | 19.303 | 0.757% | 20.8m |
| | MTPOMO | 11.823 | 1.307% | 3s | 19.656 | 2.592% | 9s |
| | MVMoE | 11.816 | 1.245% | 4s | 19.637 | 2.499% | 13s |
| | RF-POMO | 11.809 | 1.182% | 3s | 19.620 | 2.403% | 10s |
| | RF-MoE | 11.823 | 1.303% | 4s | 19.605 | 2.324% | 12s |
| | RF-TE | 11.790 | 1.027% | 2s | 19.555 | 2.062% | 9s |
| | CaDA | 11.761 | 0.779% | 2s | 19.436 | 1.441% | 9s |
| OVRPLTW | HGS-PyVRP | 10.510 | * | 10.4m | 16.926 | * | 20.8m |
| | OR-Tools | 10.497 | 0.114% | 10.4m | 17.023 | 0.728% | 20.8m |
| | MTPOMO | 10.677 | 1.572% | 2s | 17.442 | 3.020% | 9s |
| | MVMoE | 10.677 | 1.564% | 3s | 17.418 | 2.880% | 12s |
| | RF-POMO | 10.656 | 1.362% | 3s | 17.404 | 2.802% | 9s |
| | RF-MoE | 10.673 | 1.531% | 3s | 17.386 | 2.696% | 12s |
| | RF-TE | 10.646 | 1.267% | 2s | 17.328 | 2.352% | 9s |
| | CaDA | 10.613 | 0.961% | 2s | 17.226 | 1.752% | 9s |

RF-TE (1.7 M), and MTPOMO (1.3 M).

### 4.2. Testing and Hardware

We utilize the test dataset published by Routefinder (Berto et al., 2024b), which includes 1K randomly generated in-stances for each VRP variant, at scales of 50 and 100. For all neural solvers, we employ a greedy rollout strategy with ×8aug (Kool et al., 2019). This approach conducts equivalent transformations to augment the original instance and reports the best results among the eight augmented instances.

*Table 2.* Average objective function value and gap across 16 VRPs for ablation models and CaDA.

|                  | Obj.       | Gap        |
|------------------|------------|------------|
| CaDA w/o Prompt  | 11.534     | 1.926%     |
| CaDA w/o Sparse  | 11.521     | 1.795%     |
| CaDA             | **11.513** | **1.714%** |

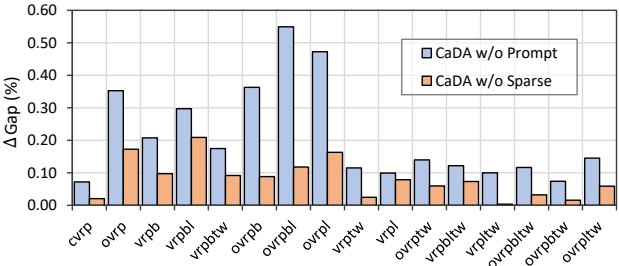

*Figure 4.* Ablation study on the proposed components of CaDA across 16 VRPs. The height of the bars represents the increased gap in performance between the model with specific components ablated and the baseline CaDA.

All experiments are run on a platform with NVIDIA GeForce RTX 3090 GPUs and Intel(R) Xeon(R) Gold 6348 CPUs at 2.60 GHz. Training our model from scratch takes about 17 hours for VRP50 and 25 hours for VRP100.

### 4.3. Main Results

In Table 1, we report the average performances for each dataset and the gaps compared to the best-performing traditional VRP solvers, as indicated by asterisks (*). Additionally, we present the total time required to solve the dataset. The best learning-based results for each dataset are highlighted with a gray background. Furthermore, we present the comparison results with the average gap on 16 VRPs for both 50-node and 100-node instances in Figure 3.

Results illustrate that the proposed CaDA method can effectively manage various VRP variants at different scales. Specifically, for VRP50 and VRP100, CaDA surpasses the second-best method by 0.26% and 0.32%, respectively. It ranks first among all neural network-based solvers for all VRP50 variants and for 13 out of 16 VRP100 variants. Similar to other neural solvers, CaDA significantly reduces the running time compared to SOTA heuristic solvers.

### 4.4. Ablation Study

In this section, we conduct ablation studies to validate the efficacy of the proposed components in CaDA. Specifically, we separately remove the constraint prompt and the Top-$k$ operation, resulting in two CaDA variants: CaDA w/o prompt and CaDA w/o Sparse, where "w/o" stands for

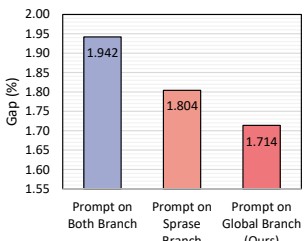 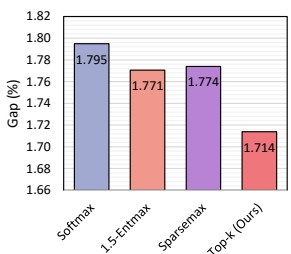

(a) CaDA with different prompt positions: concatenated to the input of the global branch, sparse branch, or both, where the first is the standard setting.

(b) CaDA with different sparse operations in the sparse branch or only standard softmax in both branches; Top-$k$ is the standard setting.

*Figure 5.* The average gap across 16 VRPs for CaDA variants under different model settings.

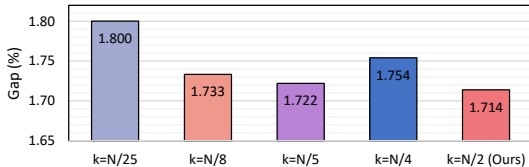

*Figure 6.* Average gap on 16 VRPs for CaDA variants with different $k$ values in the Top-$k$ selection operation, where $k = \frac{N}{2}$ is the standard setting.

"without". In CaDA w/o Sparse, both branches use the standard Softmax with global connectivity. CaDA and its variants are trained and evaluated on VRP50. During testing, ×8aug (Kool et al., 2019) is employed.

The results in Table 2 show the average gap on 16 VRPs and demonstrate that all components of CaDA make substantial contributions, with the prompt playing a particularly important role. Additionally, to study the effect of different components on various problems, we demonstrate the increased gap of these two ablation models compared with CaDA on different VRP datasets in Figure 4. These results illustrate that both the prompt and the sparse operation improve performance on all VRP variants.

Then, we further explore the influence of different CaDA settings. Specifically, using CaDA as the baseline, we conduct experiments on VRP50, focusing on the following aspects:

**Position of Prompt**   We consider three positions to introduce prompts, as shown in Figure 5(a), which illustrates the average gap across 16 VRP variants for the different models. The results indicate that integrating both sparse and prompt mechanisms within the same branch yields inferior performance, whereas concatenating the prompt only to the input of the global branch achieves the best performance.

**Different Sparse Function**   To validate the efficacy of our Top-$k$ sparse operation, we compare CaDA against three

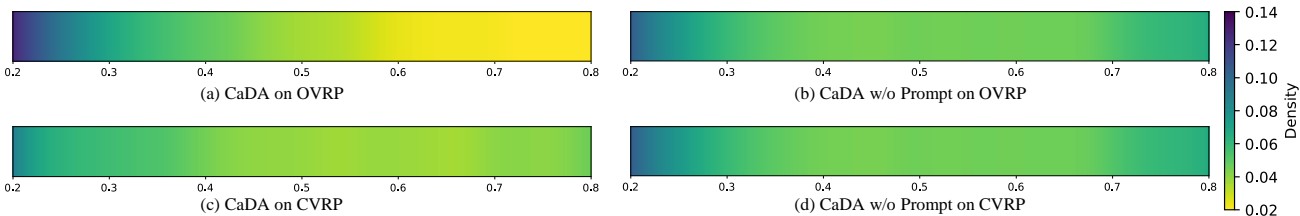

*Figure 7.* The distribution of attention weights between customers and the depot $\mathbf{A}_{i0}, i \in \{1, 2, \ldots, N\}$ for CaDA and CaDA w/o Prompt on CVRP and OVRP. Kernel density estimation (KDE) with Gaussian kernels is applied to estimate the attention weight distribution, which is visualized using a heatmap. CaDA w/o Prompt exhibits a similar attention distribution across both problems, leading to increased interference between tasks. In contrast, CaDA shows a significantly lower density of high attention values for OVRP, indicating that the proposed prompt effectively provides constraint information as depot will never be the next action for any customer in OVRP.

variants: 1) removing the Top-$k$ selection in the sparse branch, 2) replacing it with 1.5-entmax (Peters et al., 2019), and 3) replacing it with sparsemax (Martins & Astudillo, 2016). Figure 5(b) shows the average gap across 16 VRPs. Firstly, CaDA with different sparse functions consistently outperforms the version of CaDA with standard Softmax, which only has global connectivity. This indicates that the model benefits from focusing on promising nodes. Furthermore, the Top-$k$ operation outperforms the other two sparse operations, demonstrating its effectiveness for VRPs.

$k$ **for Top-$k$** To explore the different $k$ values' effects on CaDA, we conducted experiments with $k \in \left\{\frac{N}{2}, \frac{N}{4}, \frac{N}{5}, \frac{N}{8}, \frac{N}{25}\right\}$ and trained these models from scratch respectively. Figure 6 shows the average gap on 16 VRP50 datasets. It indicates that when $k = \frac{N}{2}$, the sparse branch can better cooperate with the global branch. This is the standard setting for CaDA.

### 4.5. Visualization of Constraint Awareness

To explore the influence of the prompt, we conducted further statistical experiments on the distribution of attention scores within the encoder. While the results related to the influence of the open route constraint are detailed below, the study on the influence of the time window constraint is provided in Appendix C.3. All experiments involve 100 VRP50 instances randomly selected from the test dataset, and attention scores were collected from all heads across all global layers.

**Influence on Open Route Constraint** In the case of the open route constraint, where the vehicle does not return to the depot $v_0$, the depot will never be the next node for any customer node $v_i$. Whereas in CVRP, the vehicle must return to the depot, making it a potential next node for any customer. As a result, the model should exhibit different customer-depot attention patterns for CVRP and OVRP. Specifically, in CVRP, the model should exhibit a greater number of high attention scores $\mathbf{A}_{i0}$ compared to OVRP.

Figure 7 shows the distribution of $\mathbf{A}_{i0}$ for CaDA and CaDA w/o Prompt on CVRP and OVRP. We apply KDE with Gaussian kernels to estimate the distribution of attention weights between 0.2 and 0.8. The bandwidth parameter is set to 0.1, and the KDE is visualized using a heatmap. Firstly, CaDA exhibits a significantly different distribution of attention scores between CVRP and OVRP, whereas CaDA w/o Prompt shows a similar attention distribution across both problems. When no prompt is provided, the encoder cannot distinguish between CVRP and OVRP instances, because both variants share identical input structures, with values sampled from the same distribution. This limitation leads the encoder to view instances of different variants as the same and process them with the same attention patterns.

Furthermore, when comparing CaDA on CVRP and CaDA on OVRP, we observe a significantly lower density of high attention values for OVRP, indicating that the proposed prompt effectively provides constraint information and helps the model better understand the problem.

### 4.6. Generalization to Unseen Constraints

We further conduct experiments to evaluate both zero-shot and fine-tuning performance on two unseen constraints:

- Multi-Depot (MD): Vehicles may start from any of the multiple depots but must return to their respective starting depot. Evaluation involves 16 variants, each created by adding an MD constraint to one of the 16 problems in Table 1. The number of depots is set to three, following Berto et al. (2024b).

- Mixed Backhaul (MB): Linehaul and backhaul customers can be served in any order, subject to vehicle capacity limits. We evaluate 8 variants, each formed by replacing the Backhaul constraint with the MB constraint in the 8 Backhaul-constrained problems listed in Table 1.

Table 3 shows the zero-shot performance of different mod-

*Table 3.* Zero-shot results on unseen constraints. Reported values are the average performance gap.

| Zero-shot | MD | MB |
|---|---|---|
| MTPOMO | 42.29% | 9.28% |
| MVMoE | 45.56% | 8.74% |
| RF-TE | 41.93% | 9.12% |
| CaDA | 39.34% | 8.46% |
| CaDA×32 | **28.86%** | **7.40%** |

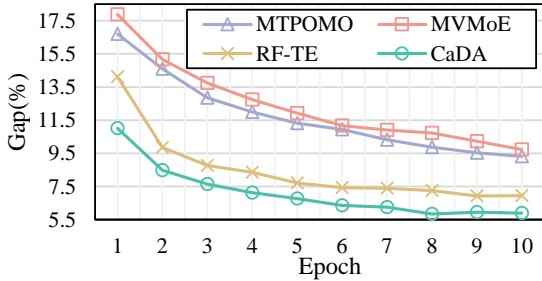

*Figure 8.* Average performance gap on 16 MD VRP variants during fine-tuning.

els on VRP50. For CaDA, we initialized the prompt using the "closest" trained task (e.g., MDOVRPBL → OVRPBL, OVRPMB → OVRPB). We additionally provide a ×32 data augmentation method for CaDA, which reports the best solution among utilizing 32 prompts. Specifically, we generate all possible combinations of a 5-dimensional binary vector $V$ (i.e., $2^5$). Given the relatively poor zero-shot performance across all evaluated models, generalization to the MD constraint appears more challenging. CaDA achieves SOTA results on both MD and MB constraints.

For the more challenging MD constraint, we further compare the fine-tuning performance of different models. We use a batch size of 128, 10,000 samples per epoch, a learning rate of $3 \times 10^{-4}$, and employ mixed-batch training. Figure 8 shows the convergence curves of each model during fine-tuning, where CaDA consistently achieves SOTA performance.

## 5. Conclusion

In this paper, we have proposed the Constraint-Aware Dual-Attention Model (CaDA), a novel cross-problem neural solver for the VRPs. CaDA incorporates a constraint prompt and a dual-attention mechanism, which consists of a global branch and a sparse branch, to efficiently generate constraint-aware node embeddings. We have thoroughly evaluated CaDA across 16 VRP variants and real-world benchmark instances. CaDA shows superior performance when compared to the current leading neural solvers. Additional ablation studies confirm the effectiveness of the proposed constraint prompt and dual-attention mechanism.

## Acknowledgements

This work was partially supported by the National Natural Science Foundation of China (Grant Nos. 62476118 and 12202472), the Natural Science Foundation of Guangdong Province (Grant No. 2024A1515011759), the Natural Science Foundation of Shenzhen (Grant No. JCYJ20220530113013031), the Guangdong Science and Technology Program (Grant No. 2024B1212010002), and the Foundation of National Key Laboratory of Aircraft Configuration Design (Grant No. ZYTS-202404).

## Impact Statement

This paper presents work whose goal is to advance the field of Machine Learning. There are many potential societal consequences of our work, none of which we feel must be specifically highlighted here.

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

# A. Related Work

## A.1. Neural Combinatorial Optimization

Neural combinatorial optimization (NCO) approaches utilize deep reinforcement learning to train a policy that constructs solutions in an autoregressive manner. Nazari et al. (2018) are the first to apply pointer networks (Vinyals et al., 2015) to solve the VRP. Subsequently, the pioneering work attention model (AM) (Kool et al., 2019) employs a powerful Transformer-based architecture. This model is optimized using the REINFORCE algorithm (Williams, 1992) with a greedy rollout baseline. Building on this, Kwon et al. (2020) introduce the policy optimization with multiple optima (POMO) method, which leverages solution symmetries and has demonstrated significantly improved performance. Subsequently, numerous studies have further refined both AM and POMO, enhancing Transformer-based methods (Xin et al., 2021; Kim et al., 2021; Peng et al., 2020; Kim et al., 2022). Given the diverse constraints and attributes in real-world transportation needs, some research focuses on various VRP variants, including heterogeneous capacitated VRP (HCVRP) (Li et al., 2022b), VRP with time windows (VRPTW) (Gao et al., 2020; Cappart et al., 2021; Zhao et al., 2021; Kool et al., 2022), and open route VRP (OVRP) (Tyasnurita et al., 2017). More information can be found in recent reviews (Bogyrbayeva et al., 2024; Li et al., 2022a).

## A.2. Cross-Problem Learning for VRPs

Neural methods for solving VRPs typically train and evaluate deep models on the same instance distributions. Some studies have explored generalization across multiple distributions (Zhang et al., 2022b; Xin et al., 2022; Geisler et al., 2022; Wang et al., 2022a; Jiang et al., 2022; Bi et al., 2022). Additionally, Zhou et al. (2023) consider both problem size and distribution variations. Recent developments have begun to address cross-problem generalization (Wang & Yu, 2023; Lin et al., 2024; Drakulic et al., 2024; Liu et al., 2024; Zhou et al., 2024a; Berto et al., 2024b; Pan et al., 2025). Wang & Yu (2023) use multi-armed bandits to achieve task scheduling. Lin et al. (2024) demonstrate how a model pre-trained on the travelling salesman problem (TSP) could be effectively adapted to targeted VRPs through efficient fine-tuning, e.g., inside tuning, side tuning, and low-rank adaptation (LoRA).

To tackle multiple VRP variants in a unified model, MTPOMO (Liu et al., 2024) conceptualizes VRP variants as combinations of underlying constraints, enabling the model to achieve zero-shot generalizability to more tasks. MVMoE proposes a new model architecture using the MoE (Shazeer et al., 2017) approach to improve performance. Furthermore, RouteFinder (Berto et al., 2024b) proposes to use a modern transformer encoder structure incorporating SwiGLU (Dauphin et al., 2017), RMSNorm (Zhang & Sennrich, 2019), and Pre-Norm (Baevski & Auli, 2019; Child et al., 2019), which considerably improves the model's capability. However, these approaches remain unaware of constraints and maintain only global connectivity throughout the encoding process, which limits their cross-problem capabilities.

## A.3. Multi-Branch Architecture

Multi-branch architectures have been widely used and have achieved success in computer vision. Some research employs multiple branches to capture both low and high-resolution image information, ultimately producing a comprehensive and powerful semantic representation that can be used for downstream tasks such as image segmentation (Ronneberger et al., 2015; Guo et al., 2022; Gu et al., 2022) or human pose estimation (Sun et al., 2019; Wang et al., 2020). Other studies assign different branches to focus on distinct aspects by utilizing attention mechanisms (Fu et al., 2019; Zhang et al., 2022a; Dong et al., 2022; Wang et al., 2023). For instance, DANet (Fu et al., 2019) proposes a dual attention network for scene segmentation, with one branch responsible for capturing pixel-to-pixel dependencies and another for capturing channel dependencies across different feature maps, thereby capturing global context (Zhang et al., 2018). Similarly, Crossformer++ (Wang et al., 2023) groups image patches in both local and global ways, incorporating short-distance and long-distance attention to achieve better representation, retaining both small-scale and large-scale features in the embeddings. Recent NCO methods adopt similar strategies for VRPs. Gao et al. (2024) propose global and local policies for CVRP and TSP, defining "local" by Euclidean distance. Fang et al. (2024) suggest learning from multiple nested local views. They both focusing on generalization across distributions and scales. In contrast, CaDA addresses 16 VRP variants using a learnable mechanism (Top-$k$ sparse attention) to dynamically select related nodes based on attention scores.

### A.4. Sparse Attention

Recent studies have proposed using sparse attention to reduce computational complexity and minimize the harmful influence of unnecessary and irrelevant items, thereby improving performance (Zhao et al., 2019; Wang et al., 2022c;b; Chen et al., 2023; Zhao et al., 2023). To achieve this, many researchers utilize pre-defined sparse attention patterns based on prior knowledge, such as local or strided attention, or combinations of multiple patterns (Guo et al., 2019; Li et al., 2019; Beltagy et al., 2020; Ainslie et al., 2020). For instance, LogSparse (Li et al., 2019) ensured that each token only attends to itself and its preceding tokens, using an exponential step size. However, these methods can be overly harsh and require well-informed prior knowledge. Another category of methods achieves sparse attention by adding an additive operation that eliminates small attention scores to exactly zero, such as the Top-$k$ operation (Zhao et al., 2019; Wang et al., 2022c;b; Chen et al., 2023) and the ReLU$^2$ operation (Zhou et al., 2024b), or employs a sparsity-inducing alternative to Softmax, such as sparsemax (Martins & Astudillo, 2016) and $\alpha$-entmax (Peters et al., 2019; Blondel et al., 2019; Correia et al., 2019). In this study, we use the simple yet efficient Top-$k$ selection operation to achieve sparse attention and enhance the representation learning from the most relevant nodes.

## B. Method Details

### B.1. MDP for Learning to Construct Solutions

When the policy autoregressively builds the VRP solution, the MDP for the decoding step $t$ can be defined as follows:

**State** $s_t \in \mathcal{S}$ is the ordered tuple $(\boldsymbol{\tau}_{t-1}, \mathcal{V})$ given by the current partial solution $\boldsymbol{\tau}_{t-1} = (\tau_1, \tau_2, \ldots, \tau_{t-1})$ and the instance $\mathcal{V}$. Initially, $\boldsymbol{\tau}_0 = \emptyset$, and at the end, $s_T$ contains a feasible solution $\boldsymbol{\tau}_T$.

**Action** $a_t \in \mathcal{A}$ is the selected index in the current step, which will be added at the end of the partial solution. If $a_t = 0$, i.e., the vehicle returns to the depot node, it signifies the end of the current sub-tour and the start of a new one.

**Policy** A neural model $\pi_\theta$ with learnable parameters $\theta$ is used as a policy to generate solutions sequentially, where the probability of generating the final feasible solution is:

$$\pi_\theta(\boldsymbol{\tau}|\mathcal{V}) = \prod_{t=1}^{T} \pi_\theta(a_t|s_t) = \prod_{t=1}^{T} \pi_\theta(\tau_t \mid \boldsymbol{\tau}_{t-1}, \mathcal{V}). \tag{17}$$

**Reward** $r \in \mathcal{R}$ can only be obtained when a whole feasible solution $\boldsymbol{\tau}_T$ is generated and is defined as the negative solution length:

$$r(\boldsymbol{\tau}_T) = -\sum_{t=1}^{T-1} d_{\tau_t \tau_{t+1}}. \tag{18}$$

### B.2. Gated Linear Unit

The standard transformer blocks include an FFD that processes the input $X$ through two learned linear projections, with a ReLU activation function applied between them. In many recent modern transformer-based large language models (Chowdhery et al., 2023; Touvron et al., 2023; Naveed et al., 2023), this configuration has been replaced by GLUs (Dauphin et al., 2017). GLUs consist of a component-wise product of two linear projections, where one projection is first passed through a nonlinear function. We employ SwiGLU (Dauphin et al., 2017), which utilizes the sigmoid linear unit (SiLU) (Elfwing et al., 2018) as the nonlinear function, as recommended in the RouteFinder(Berto et al., 2024b). The SwiGLU is defined as:

$$\text{SwiGLU}(X) = X \odot \sigma(XW_1 + b_1) \otimes \text{SiLU}(XW_2 + b_2), \tag{19}$$

where $\odot$ denotes element-wise multiplication, $\otimes$ is matrix multiplication, $\sigma$ is the sigmoid function, and $W_1, W_2, b_1, b_2$ are learnable parameters.

### B.3. Feasibility Evaluation

During decoding, at step $t$, we use the following feasibility testing procedure to identify the set of feasible actions, denoted as $I_t$.

1. Each customer node can only be visited once. If the depot is the last action in a partial solution, the next action cannot be the depot (to avoid a self-loop).

$$i \in \boldsymbol{\tau}_{1:t-1}, i \in \{1, 2, \ldots, N\} \quad \Rightarrow \quad i \notin I_t, \tag{20}$$
$$\tau_{t-1} = 0 \quad \Rightarrow \quad 0 \notin I_t. \tag{21}$$

2. For a problem without the open route constraint ($V_1 = 0$), each sub-route needs to return to the depot $v_0$ within the given limit. There are two types of constraints that enforce limits on when each sub-route must reach the depot: the time window constraint ($V_2 = 1$) with a time limit $\mathcal{T}$, and the Distance Limit constraint ($V_3 = 1$) with a distance limit $\rho$.

$$
(V_1 = 0) \wedge (V_2 = 1) \wedge \left( (z_t + d_{\tau_{t-1}i} + s_i + d_{i0}) > \mathcal{T} \right)
$$
$$
\vee \left( (V_1 = 0) \wedge (V_3 = 1) \wedge \left( l_t < d_{\tau_{t-1}i} + d_{i0} \right) \right) \Rightarrow i \notin I_t. \tag{22}
$$

3. For problems with a time window constraint ($V_2 = 1$), each customer node $v_i$ has a time window $[e_i, l_i]$ and a service time $s_i$. The vehicle must visit $v_i$ and complete its service within the specified time window.

$$(V_2 = 1) \wedge (z_t + d_{\tau_{t-1}i} + s_i > l_i) \Rightarrow i \notin I_t. \tag{23}$$

4. For the problem with the backhaul constraint ($V_4 = 1$), the backhaul will be masked if there are still linehaul services that have not been completed.

$$(\delta_i < 0) \wedge (\exists j \in \{1, 2, \ldots, N\}(\delta_j > 0) \wedge (j \notin \boldsymbol{\tau}_{1:t-1})) \Rightarrow i \notin I_t. \tag{24}$$

5. For customers, service is available when their demand does not exceed the current available capacity.

$$\left( (\delta_i > 0) \wedge (\delta_i > c_t^l) \right) \vee \left( (\delta_i < 0) \wedge (-\delta_i > c_t^b) \right) \Rightarrow i \notin I_t. \tag{25}$$

## C. Experiment Details

### C.1. Problem Setup

Table 4. 16 VRP variants with five constraints.

|  | Capacity | Open Route | Backhaul | Duration Limit | Time Window |
|---|:---:|:---:|:---:|:---:|:---:|
| CVRP | ✓ | | | | |
| OVRP | ✓ | ✓ | | | |
| VRPB | ✓ | | ✓ | | |
| VRPL | ✓ | | | ✓ | |
| VRPTW | ✓ | | | | ✓ |
| OVRPTW | ✓ | ✓ | | | ✓ |
| OVRPB | ✓ | ✓ | ✓ | | |
| OVRPL | ✓ | ✓ | | ✓ | |
| VRPBL | ✓ | | ✓ | ✓ | |
| VRPBTW | ✓ | | ✓ | | ✓ |
| VRPLTW | ✓ | | | ✓ | ✓ |
| OVRPBL | ✓ | ✓ | ✓ | ✓ | |
| OVRPBTW | ✓ | ✓ | ✓ | | ✓ |
| OVRPLTW | ✓ | ✓ | | ✓ | ✓ |
| VRPBLTW | ✓ | | ✓ | ✓ | ✓ |
| OVRPBLTW | ✓ | ✓ | ✓ | ✓ | ✓ |

In this section, we provide a detailed description of the problem setup used in this study, with the associated constraints summarized in Table 4.

**Locations**    The nodes' locations are represented by a two-dimensional vector $\vec{X}_i, i \in \{0, \ldots, N\}$, and are derived from a uniform distribution $U(0, 1)$.

**Capacity**    In this study, we consider only homogeneous vehicles, with the same vehicle capacity $C$ shared among all vehicles, and the number of vehicles is unlimited. Following the common capacity setup used in previous studies (Kool et al., 2019; Kwon et al., 2020), for $N = 50$ and $N = 100$, the vehicle capacity $C$ is set to 40 and 50 respectively.

**Node Demands**    In our study, there are two types of customers: linehaul customers with demand $\delta_i < 0$ and backhaul customers with $\delta_i > 0$ (when the backhaul constraint is active). We generate node demands as follows: we generate linehaul demands $\delta_i^l$ for all customers $i \in \{1, \ldots, N\}$ by uniform sampling from the set of integers $\{1, 2, ..., 9\}$. If the backhaul constraint is inactive, each node's true demand $\delta_i$ is equal to $\delta_i^l$. The demand generation process is now complete. Otherwise, we generate backhaul demands $\delta_i^b$ by sampling uniformly from the same set of integers $\{1, 2, ..., 9\}$. Subsequently, generate a temporary variable $y_i \sim U(0, 1)$ for each customer $i$. The demand $\delta_i$ for each customer $i$ is determined by the following rule:

$$\delta_i = \begin{cases} \delta_i^l & \text{if } y_i \geq 0.2, \\ \delta_i^b & \text{otherwise.} \end{cases} \tag{26}$$

For each node, there is a 20% probability that it represents backhaul customers in an instance.

Furthermore, before passing the demands $\delta_i$ to the policy, for training stability, we normalize the demand $\delta_i$ to the range $[0, 1]$ by $\delta_i' = \frac{\delta_i}{C}$. We set the normalized capacity to 1 to ensure that at each step of the decoding process, the remaining capacity $c_t$ also falls within the range $[0, 1]$.

**Time Windows**    For problems with time window constraints, several related factors must be considered: time windows $[e_i, l_i]$ and service times $s_i$. For the depot, $e_0 = s_0 = 0$, $l_0 = \mathcal{T} = 4.6$, where $\mathcal{T}$ represents the overall time limit for each sub-route. Additionally, the vehicle speed is 1.0.

For customers $i \in \{1, 2, \ldots, N\}$, service times $s_i$ are uniformly sampled from $[0.15, 0.18]$. Additionally, time window lengths $\Delta t_i$ are uniformly sampled from $[0.18, 0.2]$. Moreover, each customer's time window must be feasible for the tour $(0, i, 0)$; otherwise, there is no feasible tour to service this customer. Consequently, the upper bounds for the start times of the time windows are calculated as:

$$e_i^{\text{up}} = \frac{\mathcal{T} - s_i - \Delta t_i}{d_{0i}} - 1. \tag{27}$$

Subsequently, the start times of the time windows $e_i$ are determined as follows:

$$e_i = (1 + (e_i^{\text{up}} - 1) \cdot y_i) \cdot d_{0i}, \tag{28}$$

where $y_i \sim U(0, 1)$. Finally, the end times of the time windows are determined by:

$$l_i = e_i + \Delta t_i. \tag{29}$$

**Distance Limit**    For problems with the Distance Limit constraint, each sub-tour must be completed within a limit $\rho$. To ensure each instance has a feasible solution, i.e., the length of the tour $(0, i, 0)$ should remain within this limit, $\rho$ is sampled from $U(2 \cdot \max(d_{0*}), \rho_{\max})$, where $\rho_{\max} = 3.0$ is a predefined upper bound.

### C.2. Hyperparameters

The hyperparameters used for training CaDA are summarized in Table 5.

### C.3. Visualization of Constraint Awareness for Time Window Constraints

For the time window constraint, nodes $v_i$ can only be visited within their respective time windows $(e_i, l_i)$, and serving each customer costs $s_i$ time. Thus, the relationship between node pairs $(v_i, v_j)$ is influenced by their time-related factors and positional distances, i.e., $(e_i, l_i, s_i)$, $(e_j, l_j, s_j)$, and $d_{ij}$.

If an edge $(i, j)$ is legal, then it must satisfy $e_i + s_i + d_{ij} \leq l_j - s_j$. This condition means that if a vehicle starts at time $e_i$, spends $s_i$ time servicing $v_i$, and takes $d_{ij}$ time to reach $v_j$, it must arrive by $(l_j - s_j)$ at the latest to successfully service $v_j$

*Table 5.* Experiment hyperparameters.

| Hyperparameter | Value |
|---|---|
| **Model** | |
| Embedding dimension $d_h$ | 128 |
| Number of attention heads $M_h$ | 8 |
| Number of encoder layers $L$ | 6 |
| Top-$k$ | $\frac{(N)}{2}$ |
| Feedforward hidden dimension $d_a$ | 512 |
| Tanh clipping $\xi$ | 10.0 |
| **Training** | |
| Batch size | 256 |
| Train data per epoch | 100,000 |
| Optimizer | AdamW |
| Learning rate (LR) | $3e^{-4}$ |
| Weight decay | $1e^{-6}$ |
| LR scheduler | MultiStepLR |
| LR milestones | [270, 295] |
| LR gamma | 0.1 |
| Gradient clip value | 1.0 |
| Training epochs | 300 |
| Number of tasks used for training | 16 |

*Table 6.* Average gap on CVRPLIB datasets for CaDA tested with different values of $k \in \{10, 25, 50, 100\}$.

| $k$ | Obj. | Gap |
|---|---|---|
| 10 | 6440.8 | 4.61% |
| 25 | **6431.8** | **4.17%** |
| 50 | 6458.6 | 4.79% |
| 75 | 6463.7 | 4.88% |
| 100 | 6475.7 | 5.15% |

and leave the node within the service window. Accordingly, we derive the following inequality:

$$
\begin{aligned}
e_i + s_i + d_{ij} &\le l_j - s_j \\
\implies (l_j - e_i) - d_{ij} - (s_i + s_j) &\ge 0.
\end{aligned}
\tag{30}
$$

Define $\mathbf{P}_{i,j} = (l_j - e_i) - d_{ij} - (s_i + s_j)$. Figure 9 visualizes the distribution of $\mathbf{A}_{ij}$ across varying $\mathbf{P}_{ij}$. Firstly, CaDA exhibits fewer high attention values for $\mathbf{P}i, j < 0$. Additionally, for $\mathbf{P}_{ij}$ values that are too high, the vehicle may need to wait a long time at $v_j$ to wait until the start of the time window $e_j$. For example, consider $\mathbf{P}_{ij} = 4$ while the overall time limit for the sub-route is $\mathcal{T} = 4.6$. Including the edge $(i, j)$ in the solution may result in significant time wasted waiting. Compared to the CaDA w/o Prompt, CaDA also exhibits fewer high attention values for $\mathbf{P}_{ij} > 3$, indicating that CaDA more efficiently understands the problem constraints.

### C.4. Result on Real-World Instances

To further validate the effectiveness of CaDA in real-world instances, we conducted experiments using five test suites from CVRPLib[1] benchmark datasets. These datasets comprise a total of 99 instances from Sets A, B, F, P, and X (Uchoa et al., 2017), with graph scales ranging from 16 to 200, various node distributions, and customer demands. CaDA is trained on 16 different VRP types, each with a graph size of 100 nodes.

---

[1] http://vrp.atd-lab.inf.puc-rio.br/

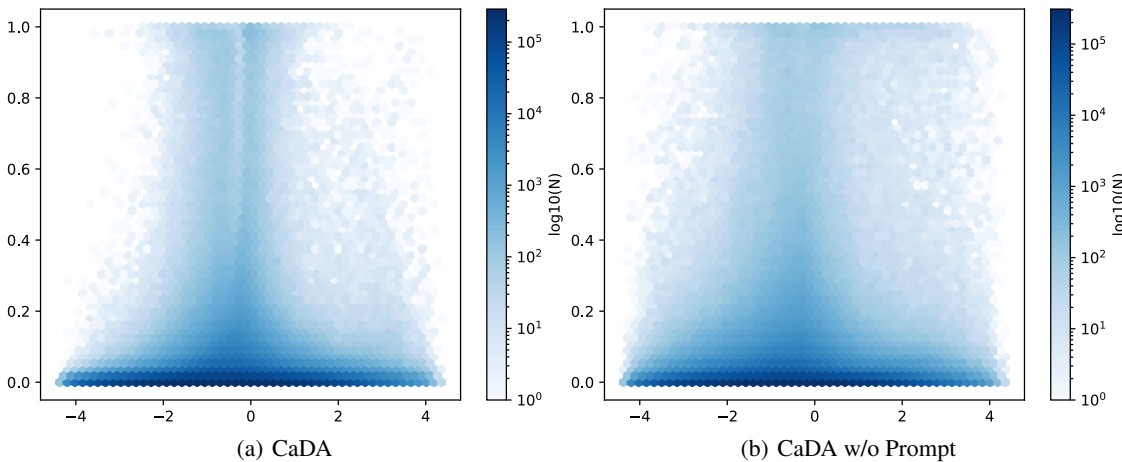

(a) CaDA                                                    (b) CaDA w/o Prompt

*Figure 9.* For 100 VRPTW instances, the distribution of attention scores $\mathbf{A}_{ij}$ across varying $\mathbf{P}_{i,j}$, where $\mathbf{A}_{ij}$ is the attention score from node $v_i$ to node $v_j$, and $\mathbf{P}_{i,j} = (l_j - e_i) - d_{ij} - (s_i + s_j)$ represents the surplus time when the vehicle starts from $v_i$ at time $e_i$ and travels to $v_j$. If $\mathbf{P}_{i,j} < 0$, the edge $(i,j)$ is illegal; if $\mathbf{P}_{i,j}$ becomes too large, including $(i,j)$ in the solution may result in the vehicle having to wait a long time for $v_j$ to open. The shade represents the density in that region. For CaDA, the attention scores at extremely high and low $\mathbf{P}_{i,j}$ values are diminished, indicating that the model successfully comprehends the time window constraint.

*Table 7.* Results on CVRPLib datasets.

|  | | MTPOMO | | MVMoE | | RF-POMO | | RF-MoE | | RF-TE | | CaDA$_{25}$ | | CaDA$_{25} \times 32$ | |
|---|---|---|---|---|---|---|---|---|---|---|---|---|---|---|---|---|
|  | Opt. | Obj. | Gap | Obj. | Gap | Obj. | Gap | Obj. | Gap | Obj. | Gap | Obj. | Gap | Obj. | Gap |
| Set A | 1041.9 | 1087.9 | 5.07% | 1071.3 | 3.07% | 1064.1 | 2.11% | 1072.2 | 2.83% | 1070.3 | 2.86% | 1069.9 | 2.76% | **1062.9** | **2.00%** |
| Set B | 963.7 | 1006.9 | 4.86% | 999.2 | 3.94% | 991.6 | 2.89% | 994.0 | 3.17% | 987.4 | 2.58% | 987.5 | 2.58% | **982.7** | **2.02%** |
| Set F | 707.7 | 820.0 | 16.23% | 791.3 | 12.16% | 804.0 | 13.93% | 813.0 | 14.31% | 794.7 | 12.95% | 748.7 | 5.74% | **745.3** | **4.99%** |
| Set P | 587.4 | 629.3 | 11.10% | 614.0 | 6.76% | 606.4 | 4.72% | 603.3 | 3.39% | 608.7 | 4.59% | 607.9 | 4.82% | **601.2** | **3.16%** |
| Set X | 27220.1 | 28952.5 | 6.09% | 28688.4 | 5.19% | 28825.4 | 5.54% | 29125.2 | 6.37% | 28520.3 | 4.46% | 28745.0 | 4.97% | **28573.9** | **4.41%** |
| Avg. | 6104.2 | 6499.3 | 8.67% | 919.8 | 5.29% | 6458.3 | 5.84% | 6521.5 | 6.01% | 6396.3 | 5.49% | 6431.8 | 4.17% | **6393.2** | **3.32%** |

We initially explore the effect of different $k$ values on the Top-$k$ sparse operation during testing. Table 6 demonstrates the average gap compared to the best-known results from CVRPLIB across all instances when testing CaDA with different $k \in \{10, 25, 50, 100\}$. The results show that when $k = 25$, CaDA achieves the best performance across different scales.

Furthermore, we compared CaDA with existing cross-problem neural solvers. Table 7 exhibits the comparison results of five test suites with the average objective function values, and the gap with the best-known solution for each dataset. The best results for each dataset are highlighted in bold. CaDA$_{25}$ represents CaDA tested with $k = 25$. Results demonstrate that the proposed CaDA$_{25} \times 32$ achieves the best performance.

