# OpenReview forum: "CaDA: Cross-Problem Routing Solver with Constraint-Aware Dual-Attention"
_ICML.cc/2025/Conference — ICML 2025 poster_

### Official Review · Reviewer_9jwN · 2025-02-20

**Overall Recommendation:** 4

**Summary:**

This paper targets cross-problem learning for vehicle routing problems. It proposes Constraint-Aware Dual-Attention (CaDA), which introduces a constraint prompt to enhance constraint awareness and employs a dual-attention mechanism consisting of a global branch and a sparse branch. The sparse branch utilizes a top-k attention strategy to focus on key node pairs. Experimental results across 16 VRP variants demonstrate the effectiveness of the proposed method.

**Claims And Evidence:**

The claims made in the submission are supported by clear and convincing evidence.

**Essential References Not Discussed:**

All relevant prior works necessary to understand the key contributions of this paper have been cited and discussed.

**Experimental Designs Or Analyses:**

The baselines and benchmark instances are sound. For the ablation experiment concerning the position of the prompt (Figure 5(a)), it would be beneficial to include an experiment that adds the prompt to both branches.

**Methods And Evaluation Criteria:**

The basic idea of the proposed CaDA is reasonable for the problem at hand, and the evaluation criteria are appropriate.

**Other Comments Or Suggestions:**

1. In Figure 2, the subsequent data flow of the sparse branch is unclear.
2. The formula in line 270 should probably be written as $\pi_{\theta}({\tau}_{t} = i \mid \mathcal{V}, \boldsymbol{\tau}\_{1:t-1})$.
3. The paper mentions that CaDA significantly reduces the runtime compared to state-of-the-art heuristic solvers. However, the results in Table 1 indicate that the runtime of CaDA is nearly identical to that of RF-TE, and it does not seem to have a significant advantage over other methods. Additionally, why do HGS-PyVRP and OR-Tools have exactly the same runtime?
4. The explanations for Figures 5(a) and 5(b) are split across two columns and are too close together, which could easily lead to confusion.
5. Please provide the specific average gap values in Figure 6.
6. It is recommended to test more data points in the ablation study on 'top-k' to better demonstrate performance variations.
7. "Kernel density estimation (KDE)" appears twice, in lines 394 and 490. It would be better to mention the full term and its abbreviation only once.
8. In Equation (19) in line 758, a comma should be used instead of a period.

**Other Strengths And Weaknesses:**

Strengths:
1. This paper demonstrates strong performance across 16 vehicle routing problems and real-world instances.
2. The paper is well-structured and easy to follow.

Weaknesses:
1. The motivation and function of the key components, e.g., the sparse branch,need further explaination.
2. The fine-tune performance is not discussed.

**Questions For Authors:**

1. Further explanation about motivation is needed. In the proposed method, the sparse branch is introduced to focus on "more related node pairs." Could you clarify why this is important?

2. How is the fine-tune performance on the VRPs.

3. In Table 2, why does CaDA w/o Sparse perform worse than CaDA? Intuitively, two standard attention branches with higher model complexity should have a stronger representation power than one standard attention branch and one sparse attention branch.

4. Some recent multi-task learning methods for NCO (e.g., [1]) are missing in the related work section.

5. In Figure 4, CaDA w/o Prompt shows a considerable performance drop on OVRPBL compared to other problems. Can the authors provide an explanation for why this happens?

[1] UniCO: On Unified Combinatorial Optimization via Problem Reduction to Matrix-Encoded General TSP. In International Conference on Learning Representations, 2025

**Relation To Broader Scientific Literature:**

The two key contributions of the paper are: it proposes the use of a constraint prompt to enhance task awareness, which has not been emphasized in existing works [1,2,3] for multi-task VRPs. Additionally, it introduces a dual-attention mechanism that incorporates a sparse attention branch to learn from more promising connections, while current methods [1, 2, 3] rely solely on standard attention mechanisms.
[1] Multi-task vehicle routing solver with mixture-of-experts. In International Conference on Machine Learning (ICML), 2024
[2] Multi-task learning for routing problem with cross-problem zero-shot generalization. In The 30th ACM SIGKDD Conference on Knowledge Discovery and Data Mining (KDD 2024). Association for Computing Machinery, 2024.
[3] Routefinder: Towards foundation models for vehicle routing problems. In ICML 2024 Workshop on Foundation Models in the Wild, 2024.

**Theoretical Claims:**

No theoretical claims are presented in this paper.

---

> ### Author Rebuttal · Authors · 2025-04-01
>
> Thank you very much for your time and effort in reviewing our work. We are very glad to know you find the paper is well-structured and easy to follow. We address your concerns as follows.
>
> > **E1. Prompt Ablation Study**
>
> To address your concern, we conduct the ablation study on adding prompt to both branches. The results below confirm that CaDA’s design (prompt on the global branch) achieves optimal performance:
>
> | Prompt on Both Branches | Prompt on Sparse Branch | CaDA  |
> | ----------------------- | ----------------------- | ----- |
> | 1.94%                   | 1.80%                   | 1.71% |
>
> > **W1,Q1. Explanation about Sparse Branch**: The motivation and function of the key components, e.g., the sparse branch,need further explaination. In the proposed method, the sparse branch is introduced to focus on "more related node pairs." Could you clarify why this is important?
>
> The sparse branch is introduced to enhance the sequential decoding process in VRP by addressing the limitations of standard attention. In VRP, selecting the next node from a small subset of nearby nodes is crucial, but standard attention assigns nonzero scores to all node pairs, diluting focus on critical decisions. The sparse branch employs Top-k sparse attention, allowing the model to concentrate on the most promising candidates based on learnable attention scores, rather than just Euclidean distances. This enables the model to automatically identify and focus on highly relevant node pairs, improving decision-making. We will add the discussion into the revised manuscript.
>
> > **W2,Q2. Fine-Tune Performance**
>
> Thank you for your valuable comment. As suggested, we conduct fine-tuning experiments, and due to the character limit, please refer to our response in E1 & Q2 to reviewer KzHG for experimental results. In summary, CaDA achieved the best zero-shot performance on two new constraints. After the first fine-tuning epoch, CaDA's gap reduced by 72% on Multi-Depot tasks (compared to 61–66% for baselines), demonstrating strong generalization capabilities.
>
> > **C1. Clarification on Sparse Branch Data Flow**
>
> The final outputs of the sparse branch are fused with those of the global branch through a fusion layer. The data from the sparse branch flows into the final node embedding $H^{(L)}$. We will revise Figure 2 to clarify this data flow.
>
> > **C2, C4, C7, C8**: The formula in line 270, The explanations for Figures 5(a) and 5(b), "Kernel density estimation (KDE)" in lines 394 and 490, Equation (19) in line 758.
>
> We appreciate your careful review. We will revise them accordingly in the revised manuscript.
>
> > **C3. Clarification on Runtime Comparisons**
>
> 1)While all neural solvers show comparable runtime performance, our key contribution is enhanced solution quality, not runtime reduction. 2) For HGS-PyVRP and OR-Tools, we set maximum runtimes (10s for VRP50, 20s for VRP100) following RF-TE (Berto et al., 2024), resulting in similar overall runtimes.
>
> > **C5, C6. Top-k Gap Values and Top-k Parameter Range**
>
> As suggested, the table below provides the specific average gaps for Figure 6, along with additional ablation studies using a wider range of $k$ values. We will include the following table in the revised manuscript:
>
> | k=2   | k=6 (N/8) | k=10  | k=12 (N/4) | k=25 (N/2) | k=40  |
> | ----- | --------- | ----- | ---------- | ---------- | ----- |
> | 1.80% | 1.73%     | 1.72% | 1.75%      | 1.71%      | 1.77% |
>
> > **Q3. Explantion about Ablation Study**
>
> Thank you for your question. Here we further explain the role of the sparse branch.
>
> - As detailed in our responses to W1 and Q1, sparse attention is beneficial for addressing the limitations of standard attention mechanisms when solving VRPs.
> - Additionally, our visualization (see Figure 4 in [PDF]([Figure.pdf](https://anonymous.4open.science/api/repo/CaDA_illustration-FC5A/file/Figure.pdf?v=13d844ac))) shows that CaDA without Sparse exhibits dispersed attention, making it difficult to focus on relevant information, whereas CaDA effectively concentrates attention on key nodes, which aligns with its better performance.
>
> >**Q4. Paper to be Cited**
>
> Thank you for your feedback. We will add this paper (UniCO, ICLR 2025) in the revised manuscript.
>
> > **Q5. Explantion about Ablation Study**
>
> Thank you for raising this point. For constraints "O", "B", and "L", it is challenging for the encoder w/o prompt to distinguish whether these constraints are "on" or "off" for a given problem instance. This is because instances with or without these constraints share identical input structures.
>
> Consequently, the encoder cannot infer which specific problem variant it is solving, leading to suboptimal node embeddings. The introduction of task-specific prompts addresses this limitation, thereby significantly improving performance.

---

> > ### Comment · Reviewer_9jwN · 2025-04-07
> >
> > The author's response has adequately addressed my concerns, and I am open to adjusting my score.

---

> > > ### Author Response · Authors · 2025-04-07
> > >
> > > Thank you very much for reviewing our response and updating the assessment. We greatly appreciate your valuable comments and feedback.

---

### Official Review · Reviewer_KzHG · 2025-03-10

**Overall Recommendation:** 3

**Summary:**

The paper proposes a novel architecture to tackle various variants of vehicle routing problems (VRP). The main idea of this architecture is to encode the different possible constraints in a so-called "constraint prompt" and use in conjunction two attention-based encoders of a VRP instance, one global and one sparse (only using top-k attention).

**Claims And Evidence:**

The authors demonstrate that their trained model can outperform other multi-task neural solvers. To make those results more conclusive, it would have been nice to discuss the following points:
- The explanation about the training process is a bit too succinct for me, since it simply refers to Berto et al. (2024). I would suggest the authors to explain/recall the training process, at least in the appendix. For instance, do the authors use mixed batch training and/or multi-task reward normalization?
- A discussion about model sizes of the different methods would help understand where the performance is coming from.
- An explanation about how the hyperparameters were obtained is missing. I believe that for the baselines, the hyperparameters suggested by their authors were used. It would be nice to confirm this point.

The authors also conduct an ablation study, which is helpful, although it seems to suggest that the sparse branch has a limited contribution. In addition, I think to make this ablation study complete, it would have been nice to test a version without the global branch.
Also, given the closeness of the results for N/2 and N/4, wouldn't it be better to use a smaller k than N/2? N/3 or N/4?

The performance of the proposed method is also confirmed on CVRPLIB. However, the best method is a bit strange to me. If I understand correctly, it consists in giving to the neural solver all possible combination of constraints to generate the constraint prompt, regardless of the constraints in effect in a solved instance. This would suggest using a similar technique when solving the 16 types of VRP in Table 1, which by construction can only improve the results. In that sense, the constraint prompt seems to encode only loosely a set of constraints.

**Essential References Not Discussed:**

Regarding the multi-branch architecture, there are some recent propositions using similar ideas (multi-view) for solving VRP, e.g.,
Gao, C., Shang, H., Xue, K., Li, D., and Qian, C. Towards generalizable neural solvers for vehicle routing problems via ensemble with transferrable local policy. arXiv, 2023
Fang, H., Song, Z., Weng, P., and Ban, Y. Invit: A generalizable routing problem solver with invariant nested view transformer. In Forty-first International Conference on Machine Learning, 2024

**Experimental Designs Or Analyses:**

The experiments are mainly conducted according to the experimental setting proposed by Berto et al. (2024). However, in contrast to previous works in multi-task solvers, the authors do not discuss much the generalization capability (e.g., one-shot or few-shot) of their method. I believe this aspect may be important in the multi-task setting.

**Methods And Evaluation Criteria:**

The proposed method uses techniques that have been proposed in other contexts (as discussed by the authors in their related work in the appendix for instance) and combine them in a somewhat novel way.

**Other Comments Or Suggestions:**

None

**Other Strengths And Weaknesses:**

The paper is quite well-written and clear, although there are a few points that could be improved in the exposition, e.g.:
- In (1), \tau_t may actually be a partial sub-tour
- Some of the architectural design decisions could be better explained in the main paper (e.g., LayerNorm in (5), or SwiGLU)
- In (14), should it be H_c^{(L)}? and W_t should be W_c?
- Below (16), what is the index g of \pi_g? Also, in the line below, u should be bolded.

**Questions For Authors:**

1. Could you clarify the training process?
2. Does the current evaluation consider generalization to new tasks?

**Relation To Broader Scientific Literature:**

I believe that the authors clearly discuss the related work, notably the recent multi-task solvers and other techniques that inspired or are related to their propositions (e.g., multi-branch, sparse attention).

**Theoretical Claims:**

There is no theoretical claim.

---

> ### Author Rebuttal · Authors · 2025-04-01
>
> Thank you for your thoughtful review of our manuscript. We are pleased to hear that you found the paper to be well-written and clear. Below, we address your concerns and questions point by point.
>
> > **E1, Q2. Generalization Capability**
>
> Thank you for raising this valuable point. We conduct experiments to evaluate zero-shot and fine-tuning performance on two unseen constraints.
>
> **Unseen Constraints:**
>
> 1. Multi-Depot (MD),  vehicles can start from any depot but must return to their respective starting depot. The evaluation includes 16 VRPs.
> 2. Mixed Backhaul (MB), linehaul and backhaul customers can be mixed along a route. The evaluation includes 8 VRPs.
>
> **Results**:
>
> |0-shot|MTPOMO|MVMoE|RF-TE|CaDA|CaDA$\times$32|
> |-|-|-|-|-|-|
> |MD|42.29%|45.56%|41.93%|39.34%|28.86%|
> |MB|9.28%|8.74%|9.12%|8.46%|7.40%|
>
> |MD\Epoch|1|5|10|
> |-|-|-|-|
> |MTPOMO|16.70%|11.32%|9.32%|
> |MVMoE|17.88%|11.92%|9.74%|
> |RF-TE|14.11%|7.71%|6.96%|
> |CaDA|11.01%|6.77%|5.90%|
>
> 1. Zero-Shot: Given the relatively poor zero-shot performance across all evaluated models, generalization to the MD constraint appears to be more challenging. For MD, CaDA achieves a gap of 39.34% (vs. 41.93–45.56% for other baselines); for MB, 8.46% (vs. 8.74–9.28%).
>
> 2. Fine-Tuning:  After the first epoch, CaDA’s gap reduces by 72% (39.34% $\to$ 11.01%), outperforming baselines’ improvements of  61–66% .
>
> We will add the results and discussion into the revised manucript.
>
> >  **C1,Q1.Training Process Clarification**
>
> 1. Mixed Batch Training: Yes, we employ this to stabilize convergence.
> 2. Reward Normalization: No reward normalization is applied.
>
> We will include these explanations in Section 4 of the revised manuscript.
>
> > **C2. Discussion of Model Sizes**
>
> MvMoE has the largest model size(3.7M), followed by CaDA(3.4M), RF-TE(1.7M), and MTPOMO(1.3M). We will add this result to the Appendix of the revised manuscript.
>
> > **C3. Hyperparameters Clarification**
>
>
> Yes, we confirm that for each baseline, the hyperparameters suggested by its authors were used.
>
> > **C4. Ablation Study About Sparse Branch**
>
> We respectfully clarify that:
>
> - In our paper, CaDA w/o Sparse retains the two-branch structure, but both use global attention.
>
> - The sparse attention consistently improved performance across all 16 VRPs (0.003–0.209\%, see Figure 4).
> - The sparse attention yields notable gains on variants such as VRPBL (0.209%), OVRP (0.173%), OVRPL (0.163%).
>
> > **C5. Removing the Global Branch**
>
> Thank you for your suggestion. We conduct ablation experiments (GA denotes global attention and SA denotes sparse attention) which show that the dual-branch model outperforms its single-branch version, and optimal performance is achieved by integrating both global and sparse attention, as in CaDA.
>
> |Branch|Attention|Gap|
> |:-|-|-|
> |Single|GA|1.92%|
> |Single|SA|1.96%|
> |Dual|GA|1.80%|
> |Dual|SA|1.75%|
> |Dual|GA+SA|1.71%|
>
> > **C6. Comparison of k = N/2 and N/4**
>
> N/2 has better results than N/4. However, choosing N/4 could result in reduced computational costs. We will include this discussion in the Section 4.4 of the revised manuscript.
>
> > **C7. CaDA$\times$32 on 16 VRPs**
>
> Thank you for this valuable point. Below, we provide the performance results of CaDA $\times  32$  on the 16 VRPs , which further improves the performance.
>
> |MTPOMO|MVMoE|RF-POMO|RF-MoE|RF-TE|CaDA|CaDA$\times$32|
> |-|-|-|-|-|-|-|
> |2.45%|2.29%|2.14%|2.16%|1.97%|1.71%|1.35%|
>
> > **R1. References to be Discussed**
>
> Thank you for your valuable suggestion. We will incorporate the papers in the revised paper. Gao et al. (2023) propose global and local policies for CVRP and TSP, defining "local" by Euclidean distance. Fang et al. (2024) suggest learning from multiple nested local views, both focusing on generalization across distributions and scales. In contrast, CaDA addresses 16 VRP variants using a learnable mechanism (Top-k sparse attention) to dynamically select related nodes based on attention scores.
>
> > **W1**:  In (1), \tau_t may be a partial sub-tour.
> >
> > **W3**:  In (14), should it be H_c^{(L)}? W_t should be W_c?
> >
> > **W4**: Below (16), what is the index g of \pi_g? u should be bolded.
>
> Thank you very much for your careful checks. We will correct these notational issues as follows:
>
> - W1: Yes. We will correct K in (1) to K_t, where K_t denotes the number of sub-tours up to the current step.
> - W3: The H_c in (14) is correct. We will correct H_c^{(L)} to H_c in (15). Yes, it should be W_c.
> - W4: Below (16), the g should be t. We will bold u.
>
> > **W2.  LayerNorm and SwiGLU Design**
>
> Thank you for the feedback. Below we clarify these design choices:
>
> - LayerNorm in (5): Since the first MLP layer's outputs vary in scale across VRPs (e.g., OVRPBLTW yields larger-scale embeddings due to more active constraints), we apply instance-level LayerNorm to normalize inputs to the second MLP.
> - SwiGLU: Following Berto et al. (2024), we use SwiGLU to improve convergence.

---

### Official Review · Reviewer_rF78 · 2025-03-12

**Overall Recommendation:** 4

**Summary:**

This paper presents Constraint-Aware Dual-Attention (CaDA), a new neural architecture for solving multi-task vehicle routing problems (VRPs). CaDA integrates a constraint prompt to help the model recognize the specific constraints of the current task, along with a dual-attention architecture that combines a standard attention branch and a top-k sparse attention branch. This architecture ensures that the encoding process is both focused on promising nodes and informed by global context. The model is evaluated on 16 different VRP variants, demonstrating significant improvements over existing neural solvers.

**Claims And Evidence:**

Yes

**Essential References Not Discussed:**

The paper has included the main related works that are crucial for understanding the context and significance of their contributions.

**Experimental Designs Or Analyses:**

I have checked all the experiments in the experimental section.

**Methods And Evaluation Criteria:**

The methods and evaluation criteria proposed in the paper are effective for addressing the intended problems.

**Other Comments Or Suggestions:**

1. Some symbols are confusing. For example, the use of $\boldsymbol{\tau}$ in line 88 might represent a solution that includes multiple sub-solutions, i.e., it is a set of tuples, and $\boldsymbol{\tau}^{i}$ represents the i-th sub-solution. However, $\boldsymbol{\tau}$ in line 152 represents a solution as a tuple, and $\boldsymbol{\tau}^{i}$ represents the i-th complete feasible solution.
2. Given that $V$ in line 163 is a vector, it might be clearer to use lowercase letters to represent it.
3. In line 261, $H_c^{(L)} $ might should be corrected to $ H_c$.

**Other Strengths And Weaknesses:**

Strengths:

1. The proposed two-branch structure is interesting.
2. It achieves SOTA results on cross-problem learning for routing problems.

Weaknesses:

There is no obvious weakness. However, some experimental results (refer to questions) require further clarification to enhance their interpretability. Additionally, since the code has not been made publicly available, the reproducibility of the experiments cannot be fully verified.

**Questions For Authors:**

1. The testing data distribution seems limited. Can CaDA achieve better performance across a broader range of distributions? For example, in [1], it provides instances with various distributions including grid, explosion, implosion, rotation, expansion, etc. It would be beneficial to test the proposed method on these varied distributions to validate its generalization capabilities.
2. Could you further explain Figure 7? Why does CaDA w/o the prompt show the same attention distribution for CVRP and OVRP?
3. The experimental setting in the "Different Sparse Functions" section (line 400) is confusing. What do you mean by "Softmax+Top-k"? Could you further explain how "a standard Softmax and a representative sparse function α-entmax" modify parts of CaDA?
4. Some experimental settings are unclear.  What is the ratio of different VRP variants in the training dataset during the training process?
5. Figure 8 needs further explanation. Why does Figure 8 appear symmetrical? And when $(i, j)$ is illegal and j will not be the next node of i, i.e., $P_{i,j} < 0$,  why are there still many $A_{ij}$ of CaDA that have large values?
[1] Towards omni-generalizable neural methods for vehicle routing problems. ICML, 2023.

**Relation To Broader Scientific Literature:**

It contributes to the community of neural combinatorial optimization. It introduces an efficient constraint prompt mechanism to improve constraint awareness. It introduces the top-k sparse operation to focus on more related node pairs.

**Theoretical Claims:**

The paper does not involve theoretical claims.

---

> ### Author Rebuttal · Authors · 2025-04-01
>
> Thank you very much for your time and effort in reviewing our work. We are very glad to know that you find our proposed method efficient and interesting. We address your concerns as follows.
>
> > **W1&Q1. Results on Other Distribution**
>
> Thank you for raising this  point. To evaluate cross-distribution generalization, we conduct additional experiments using the six distributions from [1]. The results below show that CaDA consistently outperforms all baselines across all distributions.
>
> | VRP50 | explosion | implosion | rotation | linearprojection | expansion | grid  |
> | ----- | --------- | --------- | -------- | ---------------- | --------- | ----- |
> | MTVRP | 2.65%     | 2.28%     | 2.92%    | 3.45%            | 3.02%     | 2.25% |
> | MVMoE | 2.54%     | 2.18%     | 2.77%    | 3.32%            | 2.89%     | 2.15% |
> | RF-TE | 2.25%     | 1.95%     | 2.39%    | 2.76%            | 2.59%     | 1.91% |
> | CaDA  | 1.88%     | 1.63%     | 1.97%    | 2.31%            | 2.13%     | 1.59% |
>
> > **W2. Code Availability**
>
> Thank you for your feedback. We confirm that the code will be made public immediately upon the paper's acceptance, with a link provided in the final version.
>
>
>
> > **C1-C3**: Some symbols are confusing, V in line 163, H_c^{(L)} in line 261.
>
> Thank you very much for your careful review. We will revise our notation accordingly: 1) Use distinct symbols to clearly differentiate between sub-solutions and complete solutions. 2) Change the notation from $V$ to $v$ to represent vectors. 3) Correct the symbol $H_c^{(L)}$ to $H_c$ in line 261.
>
> > **Q2. Explanation of Attention Distributions**: Could you further explain Figure 7? Why does CaDA w/o the prompt show the same attention distribution for CVRP and OVRP?
>
> Thank you for raising this point. When no task-specific prompt is provided, the encoder cannot distinguish between CVRP and OVRP instances, because both variants share identical input structures ($\mathcal{V} = \{v_i\}_{i=1}^{N}$, where $v_i = \{\vec{X}_i, A_i\}$ and $A_i = \{\delta_i\}$) , with values sampled from the same distribution. This limitation leads the encoder to view instances of different variants as the same and process them with the same attention patterns.
>
> > **Q3. Clarification of Implementation**
>
> In the "Different Sparse Functions" section, "Softmax+Top-$k$" refers to the computation of sparse attention scores ${M}( \mathbf{A} )$ in Equation (11). This involves first calculating standard attention scores via Softmax, then applying a Top-$k$ selection operation to sparsify the scores. "A standard Softmax and α-entmax" indicates replacing the Top-$k$ operation with alternative methods (e.g., α-entmax). Both approaches produce sparse attention scores but differ in their sparsification mechanisms. We will clarify this explanation in the revised manuscript.
>
> > **Q4. Training Settings**
>
> Problem variants are uniformly sampled from 16 VRPs during training (follow RouteFinder). Each batch contains a mix of variants. Therefore, each variant has roughly the same number of training instances. We will clarify this in the revised manuscript.
>
> > **Q5.Explanation of Attention Distributions**
>
> Figure 8 exhibits symmetry: larger |P_ij| corresponds to lower attention scores. This is because for P_ij > 0, a larger |P_ij| implies that including edge (i,j) in the solution might result in longer waiting times. For P_ij < 0, a larger |P_ij| indicates a smaller l_j - e_i, and node v_j's time window likely precedes v_i's, making edge (j,i) inefficient due to longer waiting times. Overall, the larger the value of |P_ij|, the less likely the two nodes should be consecutive in the solution.
>
> For cases where P_ij < 0 but |P_ij| is small, although (i,j) is illegal, (j,i) may remain feasible without excessive waiting time, so there are still many A_ij with high values.

---

### Official Review · Reviewer_BUgi · 2025-03-15

**Overall Recommendation:** 3

**Summary:**

The paper "CaDA: Cross-Problem Routing Solver with Constraint-Aware Dual-Attention" presents a novel cross-problem learning method for Vehicle Routing Problems (VRPs) that enhances constraint awareness and representation learning through a Constraint-Aware Dual-Attention Model (CaDA).

1.Main Contributions

(1) Constraint-Aware Dual-Attention Model (CaDA): A new cross-problem learning method for VRPs that improves model awareness of constraints and representation learning.
(2) Constraint Prompt and Dual-Attention Mechanism: A constraint prompt is introduced to facilitate high-quality constraint-aware learning, and a dual-attention mechanism ensures the encoding process is both selectively focused and globally informed.
(3) Superior Performance: Comprehensive evaluations across 16 VRP variants show CaDA achieves state-of-the-art performance, surpassing existing cross-problem learning methods. Ablation studies confirm the effectiveness of both the constraint prompt and dual-attention mechanism.

2. Main Results

(1) Performance: CaDA outperforms existing neural solvers (e.g., MTPOMO, MVMoE, RouteFinder) on 16 different VRP variants, with significant improvements in solution quality and efficiency.
(2) Ablation Studies: Removing the constraint prompt or sparse attention mechanism leads to performance drops, highlighting their importance. The prompt's position in the global branch and the Top-k sparse operation's effectiveness are also validated.
(3) Real-World Validation: CaDA shows strong performance on real-world CVRPLIB datasets, further proving its practical effectiveness.

3. Key Algorithm and Concepts

(1) Constraint Prompt: Represents problem constraints as a multi-hot vector processed through an MLP to generate prompts, which are concatenated with node embeddings to enhance constraint awareness.
(2) Dual-Attention Mechanism: Comprises a global branch (standard multi-head attention) and a sparse branch (Top-k sparse attention). The global branch captures broad graph information, while the sparse branch focuses on key node connections, improving representation learning.
(3) Encoding-Decoding Framework: Follows a typical encoder-decoder structure. The encoder uses the dual-attention mechanism and constraint prompt to generate node embeddings, and the decoder constructs solutions autoregressively based on these embeddings.

**Claims And Evidence:**

The claims made in this submission are well-supported by comprehensive evidence, including ablation studies, comparative experiments, and visualization analyses. The authors have addressed potential weaknesses in existing methods and provided convincing validation for their proposed approach.

1. Existing cross-problem NCO methods for VRPs are constraint-unaware and rely solely on global connectivity, limiting their performance. The authors provide a thorough review of existing methods (Section 2) and identify specific limitations in their approach to handling constraints and node relationships. This sets a clear foundation for their proposed improvements.

2. CaDA's constraint prompt and dual-attention mechanism improve cross-problem learning performance. The ablation studies in Section 4.4 demonstrate that removing either component (constraint prompt or sparse attention) results in performance degradation. This directly supports the effectiveness of both mechanisms.

3. CaDA achieves state-of-the-art results across all tested VRPs. The comprehensive experimental results in Section 4 show that CaDA outperforms existing methods on 16 different VRP variants. The results include statistical comparisons and gap percentages relative to traditional solvers.

4. The dual-attention mechanism allows the model to focus on important connections while maintaining global context. The visualization of attention weights in Section 4.5 shows distinct patterns for different VRP variants, indicating that the sparse branch effectively focuses on key connections while the global branch maintains overall context.

5. The constraint prompt effectively provides constraint information to the model. The attention distribution analysis in Section 4.5 demonstrates that CaDA with the constraint prompt exhibits different attention behaviors for different problems, while CaDA without the prompt does not. This directly supports the effectiveness of the constraint prompt.

While the evidence is generally strong, there are a few areas where additional support could strengthen the claims:

1. Computational efficiency analysis: The paper focuses primarily on solution quality but could benefit from a more detailed analysis of computational efficiency, especially for larger-scale problems beyond 100 nodes.

2. Generalization to unseen constraints: The zero-shot generalization capability to entirely new constraint combinations (beyond the 5 studied) could be further explored to demonstrate the full potential of the constraint prompt mechanism.

**Essential References Not Discussed:**

N/A

**Experimental Designs Or Analyses:**

he experimental design and analysis in this paper are robust and valid, providing convincing evidence for the effectiveness of the proposed CaDA model. The comprehensive evaluation across multiple problem variants, appropriate baseline comparisons, and insightful ablation studies all contribute to a strong experimental foundation for the claims made.

1. Experimental Setup and Design

The experimental design is comprehensive and well-structured. The authors evaluate CaDA across 16 different VRP variants with varying constraints, which demonstrates the model's versatility and generalization capabilities. The use of both 50-node and 100-node instances allows assessment of performance across different problem scales.

The experimental setup is valid for testing cross-problem learning capabilities. The inclusion of both traditional solvers (PyVRP, OR-Tools) and state-of-the-art neural solvers as baselines provides a robust comparison framework.

2. Selection and Use of Baseline Methods

The choice of baseline methods is appropriate and comprehensive. The authors compare against both traditional heuristic solvers and multiple neural approaches, including MTPOMO, MVMoE, and various RouteFinder variants.

The baselines are implemented correctly, with the authors using open-source code where available and following the same training protocols for fair comparison. The time limits for traditional solvers (10s for VRP50, 20s for VRP100) are reasonable and allow meaningful comparison with the neural methods.

3. Performance Metrics and Statistical Analysis

The primary metric, gap percentage relative to traditional solvers, is appropriate for combinatorial optimization problems. The inclusion of objective function values and running times provides a complete picture of performance.

The statistical analysis is sound. The results are presented with sufficient detail, allowing readers to assess the significance of performance differences. The use of 1K test instances per VRP variant ensures reliable performance estimates.

4. Ablation Studies

The ablation studies are well-designed and effectively isolate the contributions of the constraint prompt and dual-attention mechanisms.
The ablation studies are valid and provide clear evidence for the effectiveness of each component. The results show consistent performance improvements when both components are included, supporting the claims made.

5. Visualization and Interpretation of Results

The visualization of attention weights is insightful and supports the claims about constraint awareness.
The interpretation of results is appropriate. The authors correctly link the observed attention patterns to the expected behavior for different VRP variants, demonstrating that the model learns meaningful representations.

6. Real-World Validation

The evaluation on real-world instances from CVRPLib is a valuable addition to the experimental analysis.
The real-world validation is appropriately conducted, with results showing that CaDA generalizes well beyond synthetic datasets. The comparison with existing methods on these benchmarks further strengthens the claims.

While the experimental design is generally sound, there are a few areas where additional details or analyses could enhance the validity:

1.Statistical Significance Testing: The paper could benefit from explicit statistical significance testing between CaDA and baseline methods to quantify the confidence in performance differences.

2.Computational Efficiency Analysis: A more detailed analysis of computational resources required by CaDA compared to baseline methods would be valuable, especially for larger problem instances.

3. Generalization to Unseen Problems: While CaDA demonstrates strong performance across 16 VRP variants, testing its zero-shot generalization to completely new problem types or constraint combinations would further validate its cross-problem capabilities.

**Methods And Evaluation Criteria:**

The proposed methods in this paper, namely the Constraint-Aware Dual-Attention Model (CaDA), make excellent sense for addressing the cross-problem vehicle routing problem (VRP) challenges identified in the paper. The authors have identified key limitations in existing neural combinatorial optimization approaches for VRPs—specifically, the lack of constraint awareness and inefficient representation learning due to global connectivity—and have developed targeted solutions to these problems.

The dual-attention mechanism, combining global and sparse branches, directly addresses the need for both broad contextual understanding and focused attention on key node relationships in routing problems. This approach seems particularly well-suited to VRPs, where both global structure (like overall route efficiency) and local details (like specific customer sequences) significantly impact solution quality.

The constraint prompt mechanism effectively incorporates problem-specific information into the model, allowing it to handle diverse VRP variants without requiring separate training for each problem type. This is crucial for developing a generalizable cross-problem solver, as real-world logistics problems often involve varying combinations of constraints.

The evaluation criteria, including comprehensive testing across 16 VRP variants and comparison against both traditional solvers and state-of-the-art neural methods, provide a robust framework for assessing the model's performance. The use of gap percentage as a performance metric aligns with standard practices in combinatorial optimization and clearly demonstrates the practical significance of the improvements achieved by CaDA.

The ablation studies further strengthen the evaluation by isolating the contributions of specific model components, providing evidence for the effectiveness of both the constraint prompt and dual-attention mechanisms. These studies help establish that the proposed innovations are indeed responsible for the performance improvements observed.

Overall, the proposed methods and evaluation criteria are well-aligned with the problem objectives and demonstrate a thoughtful approach to advancing neural combinatorial optimization for VRPs.

**Other Comments Or Suggestions:**

1. Clarification of Prompt Parameters:

In Section 3.2 (Constraint Prompt), the notation for the prompt generation could be slightly clarified. The dimensions of the learnable parameters Wa, ba, Wb, and bb in Equation 5 should be explicitly stated to help readers understand the computational aspects of the prompt mechanism.

2. Practical Significance of Performance Gaps:

In Section 4.3 (Main Results), when discussing the performance gaps between CaDA and other methods, a brief discussion about the practical significance of these percentage differences in real-world logistics operations would be valuable. This could help readers better appreciate the real-world impact of the performance improvements.

3. Additional Visualizations for Ablation Study:

For the ablation study in Section 4.4, consider including additional visualizations comparing the attention patterns of CaDA with and without the sparse attention mechanism. This would provide further insight into how each component contributes to the model's performance and decision-making process.

4. Computational Efficiency Analysis:

A more detailed discussion of computational efficiency, particularly regarding how the dual-attention mechanism affects inference time compared to baseline methods, would strengthen the paper. This could include specific comparisons of runtime for different problem sizes and configurations.

5. Qualitative Analysis of Real-World Solutions:

In the real-world validation section (Appendix C.4), include a qualitative analysis of the solutions generated by CaDA for specific instances. Highlighting how the model's decisions align with expected behaviors given the constraints would provide additional confidence in its practical applicability.

Some formatting and grammar need to be standardized, for example: Page 3, Section 3.1,Page 5, Section 3.4,Page 7, Section 4.2,Page 9, Figure 5,Page 11, Section 4.5.

**Other Strengths And Weaknesses:**

Strengths:

1. Originality:

The paper demonstrates strong originality through its creative combination of constraint prompts and dual-attention mechanisms specifically tailored for cross-problem VRP solving. This represents a novel adaptation of techniques from natural language processing (prompting) and computer vision (multi-branch architectures) to combinatorial optimization.

The constraint prompt mechanism is particularly innovative in how it encodes problem-specific information to guide the model's attention, addressing a significant limitation in previous neural VRP solvers that lacked constraint awareness.

2. Significance:

The work addresses a practically significant problem with direct real-world applications in logistics and transportation. The comprehensive evaluation across 16 VRP variants demonstrates the model's versatility and potential impact across diverse routing scenarios.

The performance improvements over existing state-of-the-art methods are substantial and practically meaningful, particularly given the computational efficiency advantages of neural solvers compared to traditional heuristic approaches.

3. Clarity:

The paper is exceptionally well-written and well-structured, making complex concepts accessible to a broad audience. The methodology is clearly explained, with appropriate technical details provided in both the main text and supplementary material.

The experimental results are presented with transparency, including detailed comparisons with baseline methods, ablation studies, and visualizations that help readers understand the model's behavior.

Weaknesses:

1. Originality:

While the combination of techniques is novel, the individual components (prompting, attention mechanisms) have precedents in other domains. The paper could benefit from more extensive discussion of how this specific integration addresses limitations in previous VRP solvers beyond what has been described.

2. Significance:

The practical impact would be further strengthened by including case studies with logistics companies or real-world deployment scenarios. While the CVRPLib benchmarks are valuable, demonstrating the model's effectiveness in actual operational settings would enhance its perceived significance.

The computational efficiency analysis is somewhat limited, particularly for very large-scale problems beyond 100 nodes, which might restrict its applicability in certain high-stakes logistics scenarios.

3. Clarity:

Some sections could benefit from additional visualizations or examples to further clarify complex concepts, especially regarding how the constraint prompt interacts with the dual-attention mechanism in practice.

The paper could provide more detailed explanations of how the attention weights are visualized and interpreted, which might help readers better understand the model's decision-making process.

**Questions For Authors:**

1. Constraint Prompt Design Choices:

The constraint prompt is implemented as a multi-hot vector processed through an MLP. Why was an MLP chosen instead of other methods for incorporating constraint information (e.g., attention-based mechanisms or simple concatenation)? How sensitive is the model's performance to the specific architecture of the MLP?

If the MLP was chosen due to empirical testing showing superior performance compared to alternatives, this would strengthen the technical justification for the design. If no alternatives were tested, it might suggest that the prompt mechanism could be further optimized.

2. Dual-Attention Architecture Alternatives:

Were alternative architectures considered for combining the global and sparse attention branches (e.g., different fusion strategies or alternative attention mechanisms)? How did you settle on the current design?

If multiple alternatives were explored with the current design showing clear advantages, this would demonstrate thorough architectural search. If not, it might indicate potential for further improvement.

3. Zero-Shot Generalization to New Constraints:

Successful zero-shot generalization to entirely new constraint combinations would significantly enhance the perceived significance and versatility of the model.

Have you tested CaDA's zero-shot generalization capabilities on problem instances with combinations of constraints not seen during training? How would you expect the model to perform in such scenarios?

4. Inference Time Analysis:

How does CaDA's inference time compare to traditional solvers and other neural methods, especially for larger problem instances beyond 100 nodes? What is the computational overhead of the dual-attention mechanism compared to standard transformers?

If CaDA maintains its efficiency advantages at larger scales, this would strengthen its practical relevance. If inference time becomes prohibitive, it might limit real-world applicability.

5. Systematic Interpretation of Attention Patterns:

While the attention visualization shows different patterns for different VRP variants, is there a systematic way to interpret these patterns in terms of routing strategies, or are they primarily qualitative illustrations?

If there's a systematic interpretation method, it would demonstrate deeper understanding of the model's decision-making. If not, it might suggest a need for further interpretability research.

6. Sensitivity to Top-k Parameter:

The Top-k value is set to N/2 as the standard setting. How sensitive is the model's performance to this parameter? Have you explored a wider range of k values beyond what's shown in Figure 6?

If performance is robust across a range of k values, it suggests the mechanism is reliable. If performance is highly sensitive, it might indicate the need for careful hyperparameter tuning in practice.

7. Training Stability:

Did you encounter any training stability issues, particularly with the dual-attention mechanism and constraint prompt? How did you address them?

Evidence of stable training would support the practicality of implementing the model. If significant instability was encountered, it might indicate areas where the architecture could be refined.

**Relation To Broader Scientific Literature:**

N/A

**Theoretical Claims:**

The paper primarily focuses on empirical validation of the proposed CaDA model rather than presenting formal theoretical claims with proofs. The claims made are mostly about the effectiveness of the model architecture and its components, supported by experimental results rather than theoretical analysis:

- Comprehensive experimental results across 16 VRP variants

- Ablation studies showing the contribution of each component

- Visualization of attention patterns demonstrating constraint awareness

- Comparison against state-of-the-art methods

While the paper does not contain formal theoretical claims with proofs, the empirical evidence provided is substantial and convincing for the claims made. The authors have appropriately focused on empirical validation given the nature of the problem and the proposed solution.

---

> ### Author Rebuttal · Authors · 2025-04-01
>
> Thank you for your time and effort in reviewing our manuscript. We are glad to know that you find our method innovative and it addresses potential weaknesses in existing methods, and the paper well written and well structured. Point-to-point responses to your concerns and questions are presented below. Some visualizations are provided in the  [PDF](https://anonymous.4open.science/api/repo/CaDA_illustration-FC5A/file/Figure.pdf?v=13d844ac) .
>
> > **C1,E2,W3,S4,Q4. Computational Efficiency Analysis**
>
> We compare RF-TE (single-branch) with CaDA (dual-branch) on VRP200. CaDA remains efficient for 100+ instances. Since the encoder runs once while the decoder runs repeatedly (~100x longer), the dual-branch structure adds minimal overhead.
>
> | |Gap|Time per Instance|
> |-|-|-|
> |HGS-PyVRP| * |40s|
> |RF-TE|5.02% |0.05s|
> |CaDA|4.80% |0.05s|
>
> > **C2,E3,Q3. Generalization to Unseen Problems**
>
> We conduct a zero-shot and fine-tuning study (see E1, Q2 in KzHG for results), and CaDA achieves the best performance on two new constraints.
>
> > **E1. Statistical Significance Testing**
>
> We conduct a one-sided Wilcoxon rank-sum test comparing CaDA with RF-TE, and the result confirms CaDA’s superiority with >95% confidence.
>
> | p-value| Significant(0.05) |
> | ---| ----- |
> | 2.6E-04 | TRUE              |
>
> > **W1. How CaDA Addresses Limitations**
>
> We use prompt to enhance constraint awareness and a dual-branch model with global and sparse attention to better focus on key nodes.
>
> Our additional ablation experiments on on the position of the prompt (E1 to 9jwN) and global-sparse fusion (C1 to KzHG) confirm that our current integration is most effective. Moreover, attention visualizations (S3)  show that sparse attention reduces dispersion.
>
> > **W2. Actual Operational Settings**
>
> We validate our model on 64 real-world industrial  insatnces from MTPOMO. The table shows the average objective values.
>
> | MTPOMO | MVMoE | RF-TE | CaDA | CaDA x 32 |
> | ------ | ----- | ----- | ---- | -------------- |
> | 4262   | 4260  | 4080  | 4026 | 3983           |
>
> > **W4. Interaction Visualizations:**
>
> Figure 1 in the **PDF** illustrates the interaction mechanism: the prompt is concatenated with the initial node embeddings and fed into the global branch.
>
> > **W5. How Attention Weights are Visualized**
>
> We randomly select 100 VRP50 instances and collect global branch attention scores from CaDA and CaDA w/o Prompt. For Figure 7, KDE and a heatmap are used. For Figure 8, a hexbin plot is used.
>
> > **S1. Clarification of Parameters**
>
> The dimensions of these parameters are provided in line 194 of the manuscript, $W_a \in \mathbb{R}^{5 \times d_h}$, $b_a \in \mathbb{R}^{d_h}$, $W_b \in \mathbb{R}^{d_h \times d_h}$, and $b_b \in \mathbb{R}^{d_h}$.
>
> > **S2. Discussion of Practical Significance**
>
> CaDA outperforms the second-best learning-based methods by 0.26% for VRP50 and 0.32% for VRP100. These differences could accumulate over numerous daily routes and long-term operations, resulting in reduced transportation costs.
>
> > **S3. Ablation Visualizations for Sparse Attention**
>
> Thank you for your valuable suggestion. Figure 4 in the **PDF** shows the attention patterns: CaDA w/o Sparse exhibits dispersed attention, while CaDA effectively concentrates its attention on fewer nodes, aligning with its performance gains.
>
> >  **S5. Qualitative Analysis of Real-World Solutions**
>
> We provide a qualitative analysis in Figure 3 of the **PDF**, which shows that CaDA's solution is more similar to the best-known solution.
>
> > **S6 Standardization of Formatting and Grammar**
>
> Thank you for your suggestion. We will review the entire manuscript and standardize the formatting and grammar in the mentioned sections.
>
> > **Q1. Design of the MLP in the Prompt**
>
> 1. The multi-hot vector (dimension 5) must be projected to match the node embedding dimension. An MLP achieves this effectively. And MLPs are commonly used as prompt generators.
> 3. Ablation tests show that MLP modifications result in slight performance drops, the current design performs best.
>
> |MLP with BatchNorm|MLP w/o Norm|CaDA|
> |-- |-|-|
> |1.77%| 1.75%|1.71%|
>
> > **Q2. Fusion Strategy Design**
>
> Thank you for your question. Our fusion strategy in CaDA follows classical multi-branch architectures in computer vision. We test two alternatives：Concat and Cross-Attention. Our current design yielded the best performance.
>
> |Fusion by CrossAttn|Fusion by Concat|CaDA|
> |-|-|-|
> |1.873%|1.831%|1.714%|
>
> > **Q5. Systematic Interpretation of Attention Patterns**
>
> The attention analysis remains qualitative, but we acknowledge the need for systematic interpretation across VRPs and will highlight this as critical future work in the revision.
>
> > **Q6. Sensitivity to k Parameter**
>
> Thank you for raising this point. We test a wider range of k (refer to C5,C6 for 9jwN for results), and the model's performance is robust to k.
>
> > **Q7. Training Stability**
>
> No, the training of CaDA is stable. The loss curves for CaDA are shown in Figure 2 of the **PDF**.

---

### Decision · Program_Chairs · 2025-05-01

**Decision:**

Accept (poster)

**Comment:**

This paper proposed a learning based cross-problem solver for different VRP variants. Reviewers acknowledged the technical contributions and promising results, but also pointed out several weaknesses including relatively low novetly since it combines existing techniques and missing experimental details.